# A systematic scoping review to identify the design and assess the performance of devices for antenatal continuous fetal monitoring

Kajal K. Tamber[1]*, Dexter J. L. Hayes[1], Stephen J. Carey[2], Jayawan H. B. Wijekoon[2], Alexander E. P. Heazell[1,3]*

1 Faculty of Biology, Division of Developmental Biology and Medicine, Maternal and Fetal Health Research Centre, School of Medical Sciences, Medicine and Health, University of Manchester, St. Mary's Hospital, Manchester, United Kingdom, 2 School of Electrical and Electronic Engineering, University of Manchester, Manchester, United Kingdom, 3 Manchester University NHS Foundation Trust, St. Mary's Hospital, Manchester Academic Health Science Centre, Manchester, United Kingdom

* kajal.tamber@student.manchester.ac.uk (KKT); alexander.heazell@manchester.ac.uk (AEPH)

## Abstract

### Background

Antepartum fetal monitoring aims to assess fetal development and wellbeing throughout pregnancy. Current methods utilised in clinical practice are intermittent and only provide a 'snapshot' of fetal wellbeing, thus key signs of fetal demise could be missed. Continuous fetal monitoring (CFM) offers the potential to alleviate these issues by providing an objective and longitudinal overview of fetal status. Various CFM devices exist within literature; this review planned to provide a systematic overview of these devices, and specifically aimed to map the devices' design, performance and factors which affect this, whilst determining any gaps in development.

### Methods

A systematic search was conducted using MEDLINE, EMBASE, CINAHL, EMCARE, BNI, Cochrane Library, Web of Science and Pubmed databases. Following the deletion of duplicates, the articles' titles and abstracts were screened and suitable papers underwent a full-text assessment prior to inclusion in the review by two independent assessors.

### Results

The literature searches generated 4,885 hits from which 43 studies were included in the review. Twenty-four different devices were identified utilising four suitable CFM technologies: fetal electrocardiography, fetal phonocardiography, accelerometry and fetal vectorcardiography. The devices adopted various designs and signal processing methods. There was no common means of device performance assessment between different devices, which limited comparison. The device performance of fetal electrocardiography was reduced between 28 to 36 weeks' gestation and during high levels of maternal movement, and increased during night-time rest. Other factors, including maternal body mass index,

**Funding:** This study was funded by Tommy's from a donation from the Bateman family in memory of James Gammon to AH and JW. The funders had no role in study design, data collection and analysis, decision to publish, or preparation of the manuscript.

**Competing interests:** The authors have declared that no competing interests exist.

fetal position, recording location, uterine activity, amniotic fluid index, number of fetuses and smoking status, as well as factors which affected alternative technologies had equivocal effects and require further investigation.

## Conclusions

A variety of CFM devices have been developed, however no specific approach or design appears to be advantageous due to high levels of inter-device and intra-device variability.

## Introduction

Antepartum fetal monitoring aims to assess fetal development and wellbeing throughout pregnancy, ensuring fetuses at the highest risk of adverse outcomes are identified and appropriate investigations and interventions can be performed to mitigate the risks of fetal mortality or morbidity. In the case of chronic fetal compromise, a series of compensatory mechanisms are thought to occur in fetuses in the preceding weeks, days and hours before death, namely reductions in the fetal growth rate, progressively diminishing fetal movement (FM) and reduced fetal heart rate (FHR) variability [1,2]. These changes occur in response to fetal distress, most often caused by placental insufficiency [3,4], and can theoretically be detected by antepartum fetal monitoring. For the purpose of this review, monitoring refers to the use of multiple observational investigations over a period of time.

There are numerous predisposing risk factors for stillbirth; fetal growth restriction (FGR) is the most prominent risk factor and complicates up to 43% of fetal deaths [5]. To achieve the UK Department of Health's ambition to reduce stillbirth and neonatal mortality rates by 50% by 2030 (with a preliminary target of 20% by 2020) [6] it is important that fetuses at high-risk of mortality or morbidity are promptly identified and obtain additional monitoring. Current antenatal care guidelines provided by the National Institute of Clinical Excellence (NICE) suggest all uncomplicated pregnancies only undergo fetal growth monitoring using fortnightly symphysis-fundal height (SFH) measurements from 26–28 weeks' gestation [7]. The NHS Saving Babies' Lives Care Bundle Version 2 (SBLCBv2) [8] further suggests that pregnancies classified as high-risk for FGR have uterine artery Doppler measurement at 20–24 weeks' gestation and serial ultrasound (US) scans every 2–4 weeks in the third trimester of pregnancy to measure the estimated fetal weight and umbilical blood flow. Furthermore, if the uterine artery blood flow is normal, these serial US scans will commence from 32 weeks' gestation, and from 28 weeks' gestation if abnormal [8]. This additional monitoring ensures fetuses with an estimated fetal weight less than the tenth centile are identified in a timely manner and can receive appropriate specialist care [8].

Numerous systematic reviews have been conducted to determine the efficacy of the current forms of antenatal fetal monitoring utilised in clinical practice. In brief, neither US [9,10] and umbilical artery Doppler US [11] in low-risk cohorts, nor antenatal cardiotocography (CTG) [12] and biophysical profile [13] in high-risk cohorts, successfully reduce the stillbirth rate. Only the use of umbilical artery Doppler US in high-risk pregnancies leads to a 29% reduction in the perinatal mortality rate (risk ratio (RR) 0.71, 95% confidence interval (CI) 0.52 to 0.98, 16 studies of 10,225 participants) [14]. In addition, computerised CTG significantly reduces the perinatal mortality rate when compared to traditional CTG (RR 0.20, 95% CI 0.04 to 0.88, 2 studies of 469 participants), but this must be viewed with caution as no comparison could be made to no CTG and this review found a traditional CTG had trend to increase the perinatal

mortality rate (RR 2.05, 95% CI 0.95 to 4.42, 4 studies of 1,627 participants) [12]. These find-ings suggest the current gold-standard antenatal methods have failed to reliably detect fetal compromise. One possibility is that this is due to their intermittent nature, as these current forms of antenatal monitoring are rarely used for longer than 90 minutes, and hence only pro-vide a 'snapshot' of fetal wellbeing, thus delivering a false sense of reassurance.

Continuous fetal monitoring (CFM) offers the potential to provide an objective and longi-tudinal overview of fetal wellbeing by monitoring the fetus for prolonged periods of time (i.e. longer than 90 minutes but ideally 24 hours a day). As current methods used in clinical prac-tice do not significantly improve perinatal outcomes, hence it is hypothesised that longer-term monitoring will increase the chances that signs of fetal compromise are detected [1]. Unfortu-nately US-based technologies (US scans, Doppler and CTG) cannot be used for a sustained period of time due to concerns about the unknown thermal effects of high intensity ultrasonic waves [15]. Therefore, novel technologies to continuously monitor the fetus for prolonged periods of time have had to be developed. To date, non-invasive devices with limited safety concerns have primarily focused on monitoring the FHR or FM pattern for extended time periods. In order to implement such devices into clinical practice, the views of healthcare pro-fessionals and pregnant women must be considered, both of whom have indicate that the use of CFM devices would be acceptable [16,17].

The potential benefit of CFM devices in clinical practise is hypothetical and unproven; to date no clinical trials have compared CFM to intermittent monitoring. For such comparative trials to go ahead, CFM technology needs to be proven to be reliable. This ensures that the standard of antenatal care will not be affected by the use of CFM devices, and the hypothesis that it may improve clinical outcomes can be tested. Nonetheless, various CFM devices exist within the literature; however a clinical review of these technologies has not yet been per-formed. Therefore, this scoping review aimed to describe the available evidence to provide an overview of CFM devices developed for use in antenatal care to date, and to determine areas for improvement. This review included devices developed which can monitor the fetal heart rate and/or the pattern of FM for long periods of time. The specific aims were to: 1) describe the design and detection technology employed in the devices; 2) compare the device perfor-mance of the different CFM devices; and 3) investigate factors which affect the devices' performance.

## Methodology

### Search strategy

A scoping review was conducted using an adaptation of the methodology provided by the Joanna Briggs Institute [18]. A preliminary literature search was conducted by the primary review author (KT) using MEDLINE and Pubmed to ensure no previous scoping reviews had been conducted to assess current CFM devices developed for use in antenatal care. This also identified relevant search terms related to CFM and antenatal care.

Following preliminary searches, a systematic search strategy was developed to identify full-text articles using relevant headings (e.g. MeSH) and fields (e.g. ti,ab); adaptions were made according to the relevant databases. Eight electronic databases (MEDLINE, EMBASE, CINAHL, EMCARE, BNI, Cochrane Library, Web of Science and Pubmed) were searched by the primary review author (KT). The search strategy is described in detail in the S1 Table.

### Study screening and selection

Studies were included that described or assessed all forms of CFM during the antepartum period of pregnancy, as well as the inclusion of antepartum monitoring guidelines, where

relevant. There was no specified timeframe and articles of all languages were included; non-English papers were translated into English using a hospital- translation service. The search for guidelines was limited to national and international guidelines from the United Kingdom. Devices that were designed to monitor fetal wellbeing in labour were excluded. Further exclusion criteria were: articles not containing information relating to the aims of the review, review articles, expert opinions, commentaries, letters to the editor, or if the full-text was unable to be retrieved.

Following the removal of duplicated articles, two reviewers with clinical experience (KT and AH) independently screened all titles and abstracts to determine their eligibility with regard to the inclusion and exclusion criteria, as stated previously; the articles were then categorised into one of three groups ('included', 'excluded' or 'uncertain'). The same definition for CFM was applied as mentioned in our previous systematic review [17]; only devices which can be used in the antenatal period, are non-invasive and those which can safely be potentially used for a sustained period of time were included in this review. Ultrasound-based technologies were excluded due to heating concerns associated with prolonged use [15].

All articles in the 'included' or 'uncertain' categories underwent independent full-text assessment by two reviewer authors (KT and AH). Where disagreements between the reviewers arose, a decision was made between the reviewers following a discussion. The rationale for articles being excluded having read the full-text were recorded.

## Data extraction

A data extraction form was created and both quantitative and qualitative information was recorded from each study. Relevant information extracted from each study included the country of origin, study's aim(s), clinical context, cohort population size and characteristics, duration of fetal monitoring recordings, type of fetal monitoring device used (including the name, if relevant), the device's design and detection technology, the device performance (as reported by the authors), problems identified with the device and suggested areas for development. Data extraction was performed by the primary review author (KT), apart from information regarding the signal processing which was documented by SC.

The device performance was reported in the studies as either: (1) device accuracy, (2) signal quality (SQ) or (3) success rate. For the purpose of this review, the SQ was defined as the percentage of total recording time in which a valid FHR trace was recorded, and the success rate refers to the proportion of successful traces using pre-defined study-specific criteria for success.

## Data presentation and synthesis

Reporting of results split the devices into those which are primarily concerned with the recording FHR or FM.

For studies that presented data on the signal quality of FHR devices, meta-analysis was performed using the *metaprop* command [19] in STATA version 14 (StataCorp, TX, USA); the study signal quality and 95% confidence intervals (CI) were calculated for each study. $I^2$, a statistic derived from Cochran's chi-squared statistic Q, was calculated to describe the variability between the studies that is due to between-study variability, rather than chance [20]. An $I^2$ value of $<30\%$ was classified as low heterogeneity between studies, 30–59.9% as moderate, 60–89.9% as substantial and $\geq 90\%$ as considerable [21]. A random effects meta-analysis was used in anticipation of heterogeneity due to differences in study design and populations. When studies presented non-parametric data, estimated values of the mean and standard deviation were calculated using established methods [22], in order to enable comparison using meta-

analysis; when data were converted it has been stated in the results. Where meta-analysis was not possible, descriptive statistics were used.

## Results

The electronic search generated 4,865 hits, 22 additional studies were found by hand searching (Fig 1). Following the deletion of duplicates, 3,194 records were screened by reading their titles and abstracts. This resulted in 3,088 records being excluded, most often because they were unrelated to the topic of interest, were only applicable to intrapartum fetal monitoring or the device was unable to be used for prolonged periods of time. In total, 106 records underwent full-text review for their eligibility, resulting in 63 papers being excluded. Finally, 43 papers were included in the review. Characteristics of the included papers are shown in Table 1, and S2 Table lists the papers excluded following full-text assessment alongside their reason for exclusion.

Ten of the 43 studies were from the UK [25–27,35,37–39,49,50,55], 22 from other European countries [24,28–33,36,40,42–44,48,51–54,56,57,61,64,65], five from Japan [41,59,60,62,63], one from India [45], one from Australia [66], three from the United States of America (USA) [46,47,58], and one study from both the Netherlands and USA [34]. The included studies were heterogeneous, they had a median sample size of 34.5 participants (range 1–657) and a median number of recordings of 63 (range 1–657). The recordings had a median duration of 30 minutes (range 1–1,284 minutes); five studies did not specify the duration of recordings. The devices either recorded the FHR (35 studies) or FM pattern (8 studies).

### FHR devices

Thirty-five out of the 43 studies were specifically concerned with a device to measure the FHR. Twenty-nine studies utilised only fECG devices, of which nine were unnamed [24,25,27,30,31,40,44,49,55] and the remaining 20 studies used either: Monica AN24 device [26,33,34,36,38,39,50–52,54,56,58]; Telefetalcare [28,29,53]; FECGV1 [35]; Cardiolab Baby-card [43]; Nemo fetal monitor [48,57]; or Corometrics 112 abdominal ECG monitor [47]. Two studies used only fPCG devices, of which one was named the Fetaphon-2000 [42] and the other was unnamed [45]. Three studies investigated both fECG and fPCG devices, all of which were unnamed [32,37,41]. The remaining study used a combined fECG/fPCG device named the Invu system [46].

### Technical description

The technical details of the FHR devices are described in Table 2. With regards to fECG, there were 16 different reported device designs and technologies. Although all devices used different types and quantities of electrodes, ranging from 2–16 electrodes, all devices required a reference electrode. In addition, the arrangement of the sensors varied; all electrodes were placed abdominally, generally around the umbilicus, with the exception of four devices [32,35,41,43] which required additional thoracic electrodes. A variety of signal processing approaches were used; the most frequently described process involved the removal of the mECG trace from the device signals.

With regard to fPCG, there were five different device designs and detection technologies. All the devices used between one and four abdominal phonocardiographic sensors, and two studies stated a specific placement of the sensors—either directly over the fetal heart [37] or in a predetermined abdominal position [41]. Furthermore, two studies securely attached the sensors to the maternal abdomen using either belts [32] or a 3-D printed plastic harness [41]. All fPCG devices utilised different signal processing methods. ECG data was collected in parallel

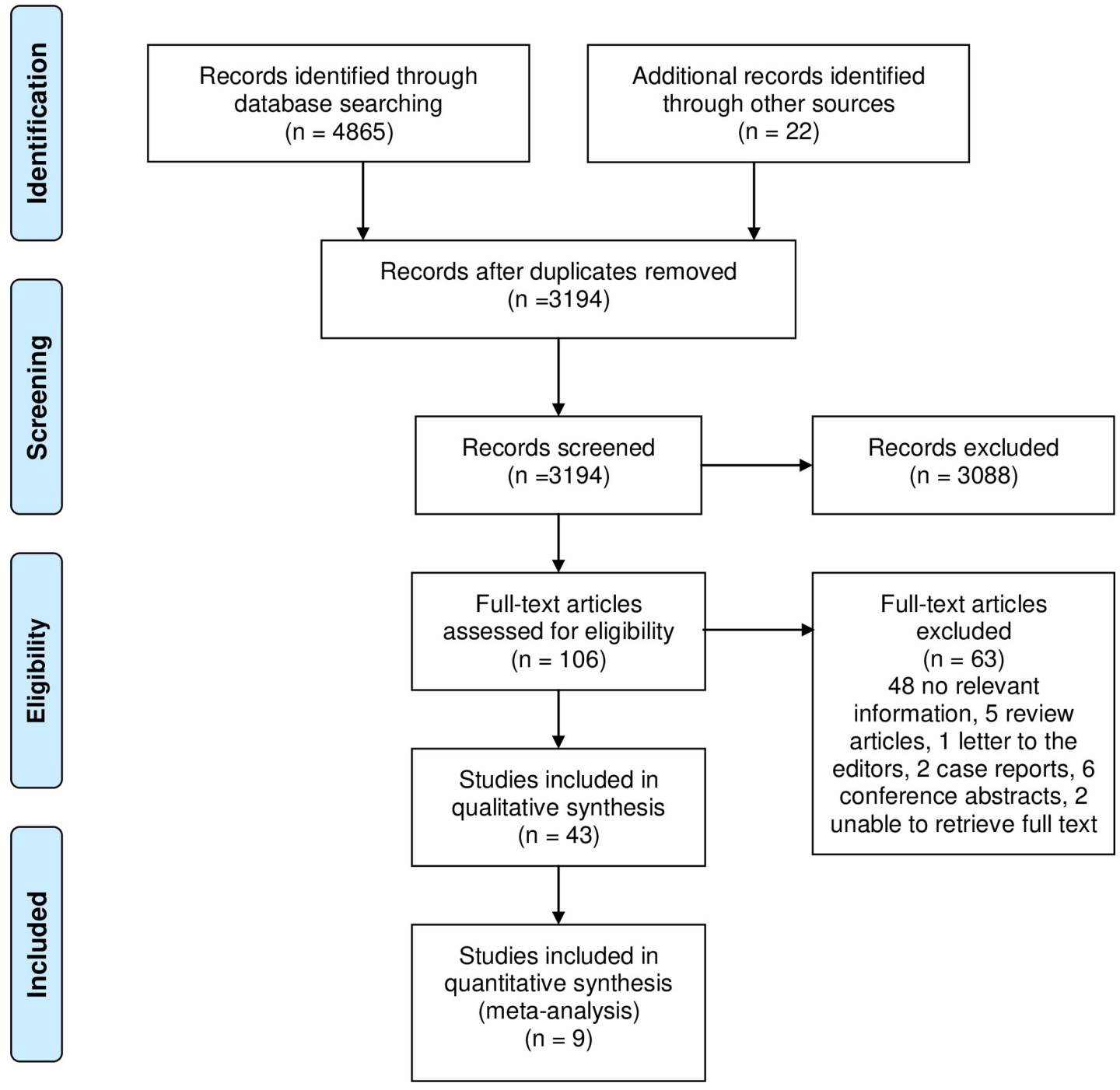

**Fig 1. PRISMA 2009 flow diagram of included studies.** Adapted from Moher et al. [23].

for three studies [32,37,40] for either validation or to aid signal recovery, with one study using a US-based Doppler instrument for reference purposes [45].The single combined fECG/fPCG device used a combination of electrodes and acoustic sensors, secured to the abdomen using a wireless belt, to monitor the FHR [46]. The FHR was independently detected by both methods and the signals combined to produce a reliable FHR trace.

**Table 1. Characteristics of included studies.**

| Authors. Year. Country. (Reference) | Device type | Device name | Type of study and clinical context | Number of participants (number of recordings) | Duration of single recording | Information of interest |
|---|---|---|---|---|---|---|
| *FHR devices* | | | | | | |
| Andreotti et al. 2017. Germany. [24] | fECG | n/a | System description and feasibility study. Assessing the device performance at University Hospital of Leipzig. | 107 (259) | 20 minutes | Device design and/or system overview. Overall device performance. |
| Carter et al. 1980. UK. [25] | fECG | n/a | Observational study. Assessing the performance of a prototype fECG module using patients at the St Mary's Hospital (London) whilst they were lying on a couch. | 56 (56) | 15 minutes | Device design and/or system overview. Overall device performance. Factors which affect fECG (gestational age). |
| Crawford et al. 2018. UK. [26] | fECG | Monica AN24 | Mixed-methods cohort study. Feasibility of wearing device overnight at home using women from St Mary's Hospital (Manchester) (tertiary maternity unit). | 22 (22) | (Median) 21 hours 24 minutes (range 11 hours 18 minutes- 23 hours 30 minutes) | Device design and/or system overview. Overall device performance. Factors which affect fECG (gestational age, BMI, time of day, maternal movement, uterine activity). |
| Crowe et al. 1996. UK. [27] | fECG | n/a | System description and feasibility study. Feasibility of obtaining fECG traces over several hours using a portable analyser. | 22 (63) | *Not mentioned.* | Device design and/or system overview. Overall device performance. Factors which affect fECG (maternal movement). |
| Fanelli et al. 2010. Italy. [28] | fECG | Telefetalcare prototype | Presentation of new device with a preliminary result. Testing of prototype for remote fetal monitoring on a single participant. | 1 (1) | *Not mentioned.* | Device design and/or system overview. Overall device performance. Factors which affect fECG (fetal position). |
| Fanelli et al. 2011. Italy. [29] | fECG | Telefetalcare | System description with preliminary results. Initial testing of device with women were sitting still in a chair. | 4 (4) | 20 minutes | Device design and/or system overview. Factors which affect fECG (maternal movement). |
| Fuchs. 2014. Poland. [30] | fECG | n/a | Observational study. Determining differences in fECG signals in uncomplicated normal and post-term pregnancies at Wroclaw Medical Academy. | 657 (657) | 30 minutes | Device design and/or system overview. Overall device performance. Factors which affect fECG (gestational age, BMI). |
| Fuchs et al. 2016. Poland. [31] | fECG | n/a | Case-control study. Clinical usefulness of fECG in comparison to CTG in normal and IUGR fetuses at Wroclaw Medical University (Poland). | 454 (454) | 30 minutes | Device design and/or system overview. Overall device performance. |
| Gobillot et al. 2018. France. [32] | fECG and fPCG | n/a | Feasibility study. Comparing the use of fECG and fPCG to CTG in normal pregnancies in Grenoble University Hospital (France). | 7 (9) | 15 minutes | Device design and/or system overview. Overall device performance. Factors which affect fECG (gestational age, maternal movement). |

*(Continued)*

**Table 1.** (*Continued*)

| Authors. Year. Country. (Reference) | Device type | Device name | Type of study and clinical context | Number of participants (number of recordings) | Duration of single recording | Information of interest |
|---|---|---|---|---|---|---|
| Graatsma et al. 2009. The Netherlands. [33] | fECG | Monica AN24 | Observational study. Device signal quality assessment at the University Medical Centre Utrecht (Utrecht) in either the home or hospital environment. | 150 (150) | 15 hours | Device design and/or system overview. Overall device performance. Factors which affect fECG (gestational age, BMI, time of day, fetal position, location of recording). |
| Graatsma et al. 2010. The Netherlands and USA. [34] | fECG | Monica AN24 | Observational study. Determining whether signal quality of fECGs is affected by maternal BMI (range of 16–50.7 kg/m$^2$) using participants from University Medical Centre Utrecht (Utrecht) and University of Maryland (Baltimore), recorded in either the home or hospital environment. | 204 (204) | 8 hours | Device design and/or system overview. Overall device performance. Factors which affect fECG (gestational age, BMI, fetal position, location of recording, maternal-fetal complications). |
| Huang et al. 1994. UK. [35] | fECG | FECGV1 | System description with feasibility study. Evaluation of a real-time fECG analysis system on women at the Queen's Medical Centre (Nottingham). | 35 (81) | *Not mentioned.* | Device design and/or system overview. Overall device performance. |
| Huhn et al. 2017. Switzerland. [36] | fECG | Monica AN24 | Case-control study. Part 1: Determining utility for antenatal assessment in healthy pregnancies and complicated pregnancies in a University Hospital. | 106 (106) | 20 minutes | Device design and/or system overview. Overall device performance. Factors which affect fECG (gestational age, BMI, time of day, maternal movement, fetal position, location of recording, maternal-fetal complications, oligohydramnios). |
| | | | Case-control study. Part 2: Determining if various factors (location, time of day, gestation, BMI etc.) affect the RQ in home recordings (uncomplicated) and hospitalised recordings (complicated). | 76 (76) | (median) 18.4 hours (IQR 14.6–22.2) | |
| Jimenez-Gonzalez et al. 2013. UK. [37] | fECG and fPCG | n/a | System description with preliminary results. Suitability of using traces estimated by SCICA (single-channel independent component analysis) from the abdominal phonogram, with comparison to reference signals. | 18 (25) | 3 or 5 minutes | Device design and/or system overview. Overall device performance. |
| Kapaya et al. 2019. UK. [38] | fECG | Monica AN24 | Observational study. Determining device signal quality in home recordings in SGA fetuses. | 35 (59) | (Mean) 17 hours and 23 minutes | Device design and/or system overview. Overall device performance. Factors which affect fECG (gestational age, time of day, smoking). |

(*Continued*)

**Table 1.** (Continued)

| Authors. Year. Country. (Reference) | Device type | Device name | Type of study and clinical context | Number of participants (number of recordings) | Duration of single recording | Information of interest |
|---|---|---|---|---|---|---|
| Kapaya et al. 2018. UK. [39] | fECG | Monica AN24 | Case-control study. Determine effect of diurnal variation, gestational age and gender on computerized CTG using fECG in SGA and normal fetuses using home recordings. | 61 (61) | 20 hours | Device design and/or system overview. Overall device performance. Factors which affect fECG (maternal-fetal complications). |
| Kariniemi et al. 1980. Finland. [40] | fECG | n/a | Feasibility study. Evaluating FHR long-term variability using an online analysis system using data from a private practise using uncomplicated pregnancies | 34 (34) | 5 minutes | Device design and/or system overview. Factors which affect fECG (gestational age). |
| Khandoker et al. 2018. Japan. [41] | fPCG and fECG | n/a | Observational study. Comparing FHR data between simultaneous fECG and fPCG obtained from women with uncomplicated pregnancies at Tohoku University Hospital. | 15 (15) | 10 minutes | Device design and/or system overview. Overall device performance. |
| Kovács et al. 2011. Hungary. [42] | fPCG | Fetaphon-2000 | System description and feasibility study. Presenting a new method of analysing fetal heart sounds in women at 34 weeks' gestation. | 225 (225) | 20 minutes | Device design and/or system overview. |
| Lakhno. 2015. Ukraine. [43] | fECG | Cardiolab Babycard | Test accuracy study. Assessing the diagnostic accuracy of fECG detecting fetal digress in normal women and women with mild/moderate pre-eclampsia and severe pre-eclampsia. | 122 (122) | 10 minutes | Device design and/or system overview. Overall device performance. |
| Manella et al. 2020. Italy. [44] | fECG | n/a | Observational study. Analysing the performance of a fECG software system using hospitalised women in the Obstetrics Unit of the University of Pisa. | 80 (80) | 5 minutes | Device design and/or system overview. Overall device performance. Factors which affect fECG (gestational age). |
| Mittra et al. 2008. India. [45] | fPCG | n/a | System description with preliminary results. Comparing a fPCG device against simultaneous US-based Doppler recordings in a hospital-based environment. | 20 (20) | 1 minute | Device design and/or system overview. Overall device performance. |
| Mhajna et al. 2020. USA. [46] | Combined fECG/fPCG device | Invu system | Observational cohort study. Comparing FHR data between a fECG/fPCG device with simultaneous CTG in a hospital-based environment. | 147 (147) | ("at least") 30 minutes | Device design and/or system overview. Overall device performance. |
| Nageotte et al. 1983. USA. [47] | fECG | Corometrics 112 abdominal ECG monitor | Observational study. Quantifying FHR variability and acceleration using fECG in complicated pregnancies at the Department of Obstetrics and Gynecology (University of California). | 188 (236) | 90 minutes | *No useable data extracted.* |

(*Continued*)

**Table 1.** (*Continued*)

| Authors. Year. Country. (Reference) | Device type | Device name | Type of study and clinical context | Number of participants (number of recordings) | Duration of single recording | Information of interest |
|---|---|---|---|---|---|---|
| Noben et al. 2019. The Netherlands. [48] | fECG | Nemo fetal monitor | Secondary analysis of a prospective cohort study. Describing the effect of betamethasone on FHR variability, by applying spectral analysis on non-invasive fECG recordings taken at the Máxima Medical Centre (Veldhoven). | 31 (124) | 30 minutes | Device design and/or system overview. Overall device performance. |
| Pieri et al. 2001. UK. [49] | fECG | n/a | System description with preliminary results. Determining success rate of FHR derived from fECG in comparison to Doppler US (as a reference) taken at the Pregnancy Assessment Centre in the Queen's Medical Centre (Nottingham). | *Not mentioned* (400) | 5–10 minutes | Device design and/or system overview. Overall device performance. Factors which affect fECG (gestational age). |
| Rauf et al. 2011. UK. [50] | fECG | Monica AN24 | Feasibility study. Determining feasibility of wearing a fECG device overnight at home using women undergoing a home induction of labour from a tertiary maternity unit (Liverpool Women's Hospital Trust, Liverpool). | 70 (70) | (Median) 10 hours 35 minutes (range 1 hour 55 minutes-22 hours 4 minutes) | Device design and/or system overview. Overall device performance. |
| Reinhard et al. 2008. Germany. [51] | fECG | Monica AN24 | Preliminary observational cohort study. Determining whether the FHR can be monitored for extended periods by fECG using hospital inpatients | 10 (10) | (Mean) 6 hours and 54 minutes (SD ± 2 hour 43 minutes) | Device design and/or system overview. Overall device performance. Factors which affect fECG (time of day). |
| Sanger et al. 2012. Germany. [52] | fECG | Monica AN24 | Observational cohort study. Evaluating fetal signal quality in fECG and comparing it to CTG. | 70 (70) | (Mean) 197.6 minutes (SD 33.2, range 116–351) | Device design and/or system overview. Overall device performance. Factors which affect fECG (gestational age, BMI). |
| Signorini et al. 2018. Italy. [53] | fECG | Telefetalcare | Presentation of new device with preliminary results. Assessing the accuracy of a prototype fECG monitor on healthy pregnant women whilst sitting in a chair at a University Medical Centre. | 5 (5) | (Mean) 30 minutes (SD 4) | Device design and/or system overview. Overall device performance. |
| Sletten et al. 2016. Norway. [54] | fECG | Monica AN24 | Observational study. Assessing the feasibility of long-term fECG and studying the FHR in the home environment in healthy pregnancies. | 12 (12) | (Median) 18.8 hours (range 17.4–19.3) | Device design and/or system overview. Overall device performance. Factors which affect fECG (BMI). |
| Taylor et al. 2003. UK. [55] | fECG | n/a | Cross-sectional observational study. Determining a fECG system's ability to acquire separate FHR signals recordings in singleton, twin and triplet pregnancies | 304 (381) | 5 minutes | Device design and/or system overview. Overall device performance. Factors which affect fECG (gestational age, multiple fetuses). |

(*Continued*)

**Table 1.** (Continued)

| Authors. Year. Country. (Reference) | Device type | Device name | Type of study and clinical context | Number of participants (number of recordings) | Duration of single recording | Information of interest |
|---|---|---|---|---|---|---|
| Van Leeuwen et al. 2014. Germany. [56] | fECG | Monica AN24 | Observational study. Part 1: assessing signal availability in women wore the monitor for prolonged periods of time, and were allowed to move. | 63 (130) | (Mean) 301 minutes ± 212 (range 21–1048) | Device design and/or system overview. Overall device performance. Factors which affect fECG (gestational age, BMI, fetal position, maternal-fetal complications). |
| | | | Test accuracy study. Part 2: determining accuracy of beat-to-beat detection of fECG in women attending Marienhospital (Witten) in supine/semi-supine position (with no movement) compared to a different historical control group who had fMCG recordings. | 51 (55) | (Mean) 51.4 minutes ± 10.5 (range 30.5–70.1) | |
| Verdurmen et al. 2018. The Netherlands. [57] | fECG | Nemo fetal monitor | Prospective cohort study. Describing the effect of betamethasone on FHR variability, by applying spectral analysis on non-invasive fECG recordings taken at the Máxima Medical Centre (Veldhoven). | 28 (122) | *Approximately* 30 minutes. | Device design and/or system overview. Overall device performance. |
| Yilmaz et al. 2016. USA. [58] | fECG | Monica AN24 | Observational cohort study. Comparing fECG intervals across gestation in fetuses with (n = 51) and without (n = 41) structural cardiac defects at a University Medical Centre. | 92 (177) | 45 minutes | Device design and/or system overview. Overall device performance. Factors which affect fECG (gestational age, maternal-fetal complications). |
| *FM devices* | | | | | | |
| Kamata et al. 2017. Japan. [59] | Accelerometer | FMAM | Observational study. Determining the degree of fetal hiccup occurrence using weekly/biweekly overnight recordings at home. | 23 (174) | Mean 6.21 hours (SD ± 0.96 hours) | Device design and/or system overview. Overall device performance. |
| Mesbah et al. 2011. Australia. [44] | Accelerometer | AFAM | System description with preliminary results. Determining device performance at detecting fetal movement in comparison to real-time US at the Royal Brisbane and Women Hospital, Brisbane. | 3 (3) | *Not mentioned.* | Device design and/or system overview. Overall device performance. Factors which affect performance of accelerometers (gestational age). |
| Nishihara et al. 2015. Japan. [60] | Accelerometer | FMAM | Test accuracy study. Part 1: comparing fetal movement detected by the FMAM device with those manually recorded between 28–38 weeks' gestation. | 6 (44) | 30 minutes | Device design and/or system overview. Overall device performance. Factors which affect performance of accelerometers (gestational age). |
| | | | Observational cohort study. Part 2: establishing a new fetal movement count index using overnight home recordings taken by women once every 4 weeks. | 12 (44) | 30 minutes | |

(*Continued*)

**Table 1.** (Continued)

| Authors. Year. Country. (Reference) | Device type | Device name | Type of study and clinical context | Number of participants (number of recordings) | Duration of single recording | Information of interest |
|---|---|---|---|---|---|---|
| Rooijakkers et al. 2014. The Netherlands. [61] | fVCG | n/a | System description with preliminary results. Quantification of fetal movements using fECG in comparison to simultaneous US at the Máxima Medical Centre (Veldhoven). | 4 (4) | 30 minutes | Device design and/or system overview. Overall device performance. |
| Ryo et al. 2018. Japan. [62] | Accelerometer | FMAM | Observational cohort study. Obtaining normal reference values for fetal movement, using overnight home recordings taken weekly by women from 28 weeks' gestation until term. | 64 (385) | (Mean) 6.39 hours (SD 1.2) | Device design and/or system overview. Overall device performance. |
| Ryo et al. 2012. Japan. [63] | Accelerometer | FMAM | Preliminary observational cohort study. Part 1: determining accuracy of recording fetal movements at rest in uncomplicated pregnancies | 14 (45) | 30 minutes | Device design and/or system overview. Overall device performance. Factors which affect performance of accelerometers (gestational age, position of sensor). |
| | | | Preliminary observational cohort study. Part 2: feasibility of using a fetal movement device numerous times (as often as they could) in a home environment overnight. | 6 (53) | Overnight (*specific duration not mentioned*) | |
| Vullings et al. 2008. The Netherlands. [64] | fVCG | n/a | Feasibility study. Initial testing of using fVCG to detect fetal movement, with comparison to simultaneous US in a single healthy patient. | 1 (1) | (*"at least"*) 5 minutes | Device design and/or system overview. |
| Vullings et al. 2013. The Netherlands. [65] | fVCG | n/a | System description and performance analysis. Describing the maximum a posteriori (MAP) method for fVCG loop alignment and comparing the fVCG device performance against simultaneous US in recordings taken at the Máxima Medical Centre (Veldhoven). | 8 (8) | 10–20 minutes | Device design and/or system overview. Overall device performance. |

AFAM: Accelerometer-based fetal activity monitor. BMI: Body mass index (kg/m$^2$). CTG: Cardiotocography. fECG: Fetal electrocardiogram. FHR: Fetal heart rate. FMAM: Fetal movement acceleration measurement. fPCG: Fetal phonocardiography. fVCG: Fetal vectorcardiography. IUGR: Intrauterine growth restriction. MHR: Maternal heart rate. n/a: Not applicable. RQ: Recording quality. SD: Standard deviation of mean. SGA: Small-for-gestational age. UK: United Kingdom. US: Ultrasound. USA: United States of America.

**Performance of fECG devices.** There was no common means of assessment to determine the accuracy or success rates of the devices in the included studies. Therefore, the accuracy or success of different named devices is not directly comparable; however studies which used the same named device were compared.

Eight out of twelve studies utilising the Monica AN24 device reported the SQ and the variability. The mean SQ was 68% (95% CI 48–87%), however there was considerable heterogeneity between the studies ($I^2$ = 97.94%) (Fig 2). There was no significant relationship between

**Table 2. Fetal heart rate device descriptions.**

| Device name | Type of sensor/ electrode | Number of sensors/ electrodes | Arrangement of sensors/ electrodes | Placement of reference electrode | Signal processing | Additional information |
|---|---|---|---|---|---|---|
| *fECG devices* | | | | | | |
| Monica AN24 [26, 33, 34, 36, 38, 39, 50–52, 54, 56, 58] | Proprietary 5-electrode patch. | 4 electrodes with 1 "ground" electrode. | 2 electrodes along the midline (at the side of the uterine fundus & above the symphysis), & 1 at each side of the uterus. | Left flank. | Proprietary. | Able to record FHR, MHR, uterine activity and level of maternal movement. Skin preparation (exfoliation) was performed prior to the application of electrodes in 6 studies [26,33,36,50,54,56]. |
| Telefetalcare [28,29,53] | Woven silver textile electrodes. | 8 electrodes and 1 reference electrode. | Around the umbilicus in a circle. | At the umbilicus. | Field Programmable Gate Array (FPGA)-based principle component analysis (PCA). | Able to detect both FHR & MHR. Composed of 2 units: (1) wearable unit–a bodysuit made of cotton and Lycra and the recording device; (2) a dock for transmission of data through the telephone line or a smartphone/tablet that sends signals over the network to a remote diagnostic centre and receives their results. |
| FECGV1 [35] | *Not mentioned.* | *Not mentioned.* | Attached to thorax and abdomen (specific placement not mentioned). | *Not mentioned.* | mECG detection, followed by cancellation from abdomen ECG. | n/a |
| Cardiolab Babycard [43] | *Not mentioned.* | 6 electrodes with 1 common reference and 1 active ground. | 5 electrodes placed in a semi-circle around the top of the abdomen (FHR), 1 electrode placed on the left chest (MHR) and 1 electrode placed on the right chest (active ground). | Symphysis pubis. | Pre-filtering, mECG detection, periodic component analysis and mECG cancellation, wavelet filter, inverse periodic component analysis, fECG detection. | Wireless. |
| Nemo fetal monitor [48,57] | *Not mentioned.* | 6 electrodes with 1 reference electrode and 1 ground electrode. | 6 electrodes placed in a circle around the umbilicus with the ground electrodes placed at the umbilicus. | At the umbilicus. | Sampling rate of 500 Hz. mECG detection, spatial combination to enhance SNR, fetal R peak detection. | n/a |
| (*Unnamed Andreotti fECG device*) [24] | Kendall™ ARBO H98SG. | 8 (1 maternal chest lead and 7 abdominal channels) and 1 reference electrode. | 4 external derivations (forming a larger circle around the maternal abdomen) and 3 internal (around the umbilicus). | About the fundus of the uterus. | Bayes classifier and adaptive Kalman filter. | The electrodes were coupled to ADInstruments hardware. |
| (*Unnamed Carter fECG device*) [25] | Welsh electrodes (Bowen and Company Inc.) | 2 with leg plate as reference. | 1 placed at the uterine fundus, and 1 placed above the symphysis pubis. Electrodes rearranged if no signal. | On the leg. | Manual. The blanking method of signal processing is utilised by the Sonicaid FM3R abdominal ECG module. | Electrodes coupled to the skin with Redux cream (Hewlett Packard Ltd.) |
| (*Unnamed Crowe fECG device*) [27] | *Not mentioned.* | *Not mentioned.* | Vertical placement of electrodes with a separation of 20–25 cm in line with and equidistant from the umbilicus. | *Not mentioned.* | Signal processing using Texas Instruments 32010 digital signal processor (DSP) (a 16 bit, integer arithmetic device). Then simple averaging and subtraction of maternal and fetal signals with matched filtering. | Skin cleaned before electrode application. |

(*Continued*)

**Table 2.** (Continued)

| Device name | Type of sensor/ electrode | Number of sensors/ electrodes | Arrangement of sensors/ electrodes | Placement of reference electrode | Signal processing | Additional information |
|---|---|---|---|---|---|---|
| (*Unnamed Fuchs fECG device*) [30,31] | Disposable electrodes 3M type 2222. | 5 electrodes and 1 reference electrode. | 1. 5cm right of umbilicus 2. 10cm right of umbilicus 3. 5cm above umbilicus 4. 1cm left of umbilicus 5. 10 cm below the inguinal region on the front side of thigh (return electrode). | 10 cm below umbilicus. | Signals amplified using a remote amplifier box, then analysed and stored using the KOMPOREL software. | Skin prepared with mild abrasion using sand paper material at the site of electrode placement. Additional gel layer applied on sensing element. |
| (*Unnamed Gobillot fECG device*) [32] | BioAmp enhancer, ADInstruments. | 6 electrodes with 2 reference electrodes. | 2 thoracic electrodes (for MHR) and 4 abdominal electrodes (for FHR) (specific placement not mentioned). | 1 thoracic and 1 abdominal (specific placement not mentioned). | fECG established from abdominal ECG and an adaptive kernel filtered maternal ECG. | n/a |
| (*Unnamed Jimenez-Gonzalez fECG device*) [37] | "Plate" electrodes. | 3 plates with 1 reference plate. | Placed on the maternal womb to form an equilateral triangle (between 20 and 25 cm side). | Placed close to the right ankle. | Electrodes connected to a lead selector-ECG (PB-640G, Nihon Kohden™) and then to an instrumentation amplifier (AB-621G, Nihon Kohden™) for conditioning purposes. BP filtered at 10-40Hz with mix auto and manual beat marking. | n/a |
| (*Unnamed Kariniemi fECG device*) [40] | *Not mentioned.* | *Not mentioned.* | *Not mentioned.* | *Not mentioned.* | Micro-processor-based system which utilises the QRS detection pulses of a cardiotocograph. | n/a |
| (*Unnamed Khandoker fECG device*) [41] | *Not mentioned.* | 11 electrodes with 1 reference electrode. | 10 placed on abdomen (specific placement not mentioned) and 1 in the right thoracic region. | On the back (specific placement not mentioned). | Data acquisition system (IRIS™, Atom Medical Co. Japan) with 1000 Hz sample rate and 16 bit resolution. Processed via mECG cancellation and blind source separation. | n/a |
| (*Unnamed Manella fECG device*) [44] | Ambu BlueSensor T electrodes. | 9 electrodes and 1 ground electrode. | Left (L), Right (R) and Foot (F) electrodes were placed in a triangle around the umbilicus, and the precordial ones (C1, C2, C3, C4, C5, C6) were put an external circle around this triangle. | On the right side of the patient. | Basic filtering followed by independent component analysis (ICA) to find the mECG, then mECG removed by SVD. | Standard 12 leads ECG of the Cardioline S.p.a." company. |
| (*Unnamed Pieri fECG device*) [49] | *Not mentioned.* | 3 electrodes with 1 reference electrode. | Shown on a diagram as roughly: (1) 3cm above the umbilicus and 5cm to the left (2) 5cm above the umbilicus (3) 3cm above the umbilicus and 5cm to the right. | Symphysis pubis. | mECG detection via matched filter and mECG suppression via subtraction. | n/a |

(*Continued*)

**Table 2.**  (Continued)

| Device name | Type of sensor/ electrode | Number of sensors/ electrodes | Arrangement of sensors/ electrodes | Placement of reference electrode | Signal processing | Additional information |
|---|---|---|---|---|---|---|
| (*Unnamed Taylor fECG device*) [55] | Glued electrodes. | 12 or 16 electrodes with 1 reference electrode. | Evenly placed over the whole abdominal wall. | Right ankle. | A QinetiQ box was used for signal processing. | Skin was prepared for low impedance by gentle excoriation of surface skin cells (3M Skinprep 2236). |
| *fPCG devices* | | | | | | |
| Fetaphon-2000 [42] | *Not mentioned.* | 2 sensors. | *Not mentioned.* | n/a | Sampled at 333Hz and BP filtered from 25-200Hz with an active filter. | Able to record FHR and uterine contractions. Made up of 2 parts: (1) home monitor and (2) a system consisting of a mobile phone network & the Internet as transmitting elements & the Evaluation Centre. |
| (*Unnamed Gobillot fPCG device*) [32] | Cardio-microphones (MLT201). | 2 microphones. | *Not mentioned.* | n/a | Sampled at 1kHz and BP filtered from 20Hz to 250Hz followed by envelope detection. | Held on the skin by belts. The position of the fetus was checked so the sensors could be placed in optimal positions. |
| (*Unnamed Jimenez-Gonzalez fPCG device*) [37] | TK-701T, Nihon Kohden™. | 1 fPCG piezoelectric transducer. | Placed on the maternal abdomen, close to the fetal heart (found prior to placement using US). | n/a | Transducers connected to ADI amplifier; 500Hz rate; Hilbert transform envelope detection, followed by 4-11Hz FIR filter. | n/a |
| (*Unnamed Khandoker fPCG device*) [41] | fPCG sensors. | 4 sensors. | All sensors placed equidistant from the umbilicus. | n/a. | Decomposition technique: source separation Algorithm and denoising + Notch filter (50Hz). 4 channels were amplified & digitized by Powerlab 26T data acquisition system (ADInstruments Inc) & recorded by a laptop computer. | The sensors were embedded in a high definition 3D printed plastic harness. |
| (*Unnamed Mittra fPCG device*) [45] | Piezoelectric sensor. | 1 sensor with 1 reference sensor. | On abdomen (specific placement not mentioned but assumed to be over fetal heart). | In open air next to signal sensor. | Noise cancelled from reference with adaptive filter; BP filter 35-200Hz, followed by envelope detection, normalisation & thresholding for FHR extraction. | n/a |
| *Combined fECG/fPCG device* | | | | | | |
| Invu system [46] | Not mentioned | 8 electric sensors and 4 acoustic sensors. | Half of the electric/ acoustic sensors placed on upper half of abdomen, and the remained on the lower half of the abdomen using 2 belt straps. | *Not mentioned.* | Data acquired at a sample rate of 250 Hz, digitized, and sent via Bluetooth to a mobile device for analysis by an algorithm on cloud-based servers. The algorithm validates the data, pre-processes the data to remove noise, detects FHR independently from the 2 data sources (electrical and acoustic), and fuses the detected heartbeat arrays to calculate FHR. | Able to record FHR and MHR. The Invu system is wearable; the wireless belt and the sensors remain in a fixed location. The data can be accessed by both the pregnant women or healthcare provide by a mobile phone application. |

The Corometrics 112 abdominal ECG monitor [47] was not included in this table as there was no usable information provided.

BP: Band pass. fECG: Fetal electrocardiography. FHR: Fetal heart rate. fPCG: Fetal phonocardiography. mECG: Maternal electrocardiography. MHR: Maternal heart rate. n/a: Not applicable. SVD: Singular value decomposition. US: Ultrasound.

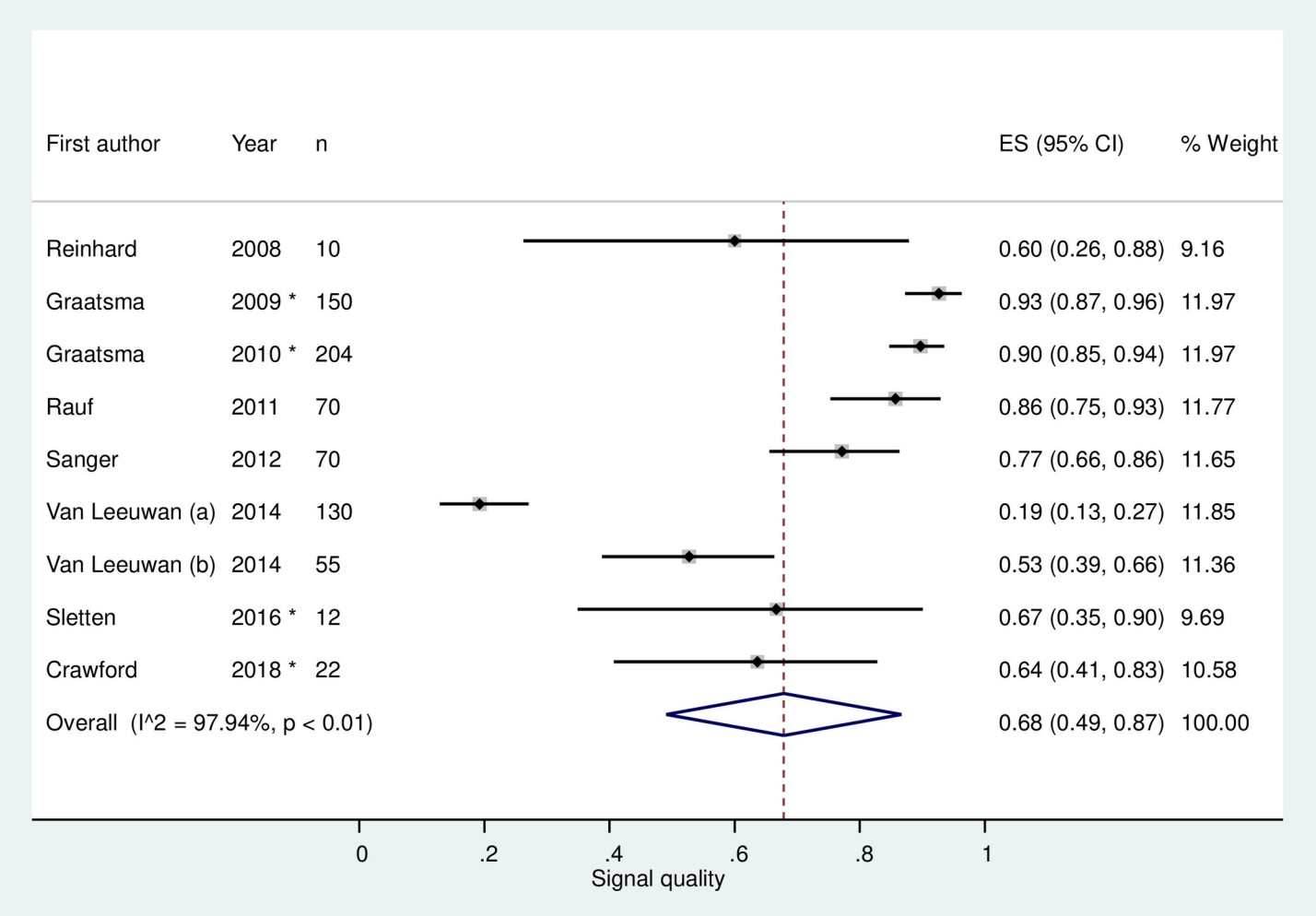

**Fig 2. Estimated average signal quality of fetal electrocardiography recordings from eight studies which used the Monica AN24 device.** Black markers represent the signal quality with 95% confidence intervals (CI) (whiskers). The size of each grey square represents the relative weight in the meta-analysis. The diamond represents the signal quality summary value. Asterisk (*) represents studies which had the median and interquartile range/range data converted to the mean and standard deviation.

study size and the estimated SQ (p = 0.53, $r^2$ = 0.06). A sub-group analysis was performed to look at the effect of converted data on heterogeneity, however no significant effect was seen (P = 0.093) between the studies with unconverted data (mean SQ 86%, 95% CI 79–94%) and those with converted data (mean SQ 59%, 95% CI 28–90%). The remaining four studies could not be included in the meta-analysis as they either did not provide an overall SQ [36,58] or they did not provide all of the required data (e.g. SD or IQR) [38,39].

Both Telefetalcare studies [29,53] reported identical values for the accuracy (91.3%) and sensitivity (92.9%) of the device at detecting fetal QRS complexes. The prototype of the Telefetalcare device had a reported accuracy of 95% following testing on a single participant [28]. The FECGV1 device detected 72% of fECG complexes (P and QRS waves) overall [35]. The Cardiolab Babycard device had both a sensitivity and specificity of 100% when detecting fetal distress in women suffering from pre-eclampsia [43].

Ten studies which utilised unnamed fECG devices described the proportion of successful traces using pre-defined criteria for success (Fig 3). The criterion was either: traces being

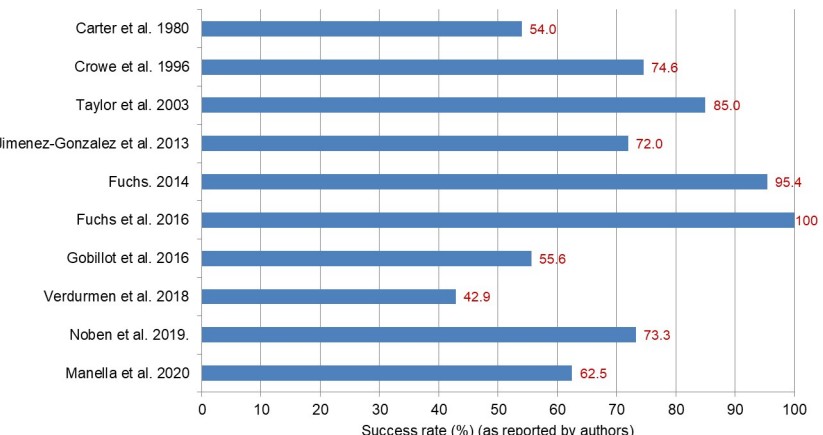

**Fig 3. Success rates of unnamed fetal electrocardiography devices.** Success rates presented as reported by the authors.

above a certain level of quality [25,27,31,44,48,57]; traces with successful signal separation [55]; traces with successful estimation of the FHR [32,37]; or traces with successful measurement of the T/QRS ratio [30]. The success rates have a broad range (42.9–100%), and there is no obvious association between the year of the study and the success indicating no obvious improvement over time.

**Performance of fPCG devices.** The accuracy of fPCG was measured by four out of five studies using reference signals obtained from additional devices. Two studies [32,45] used US-based technologies as a comparison; one reported correlations of 75% and 80% (n = 2) between fPCG and CTG FHR estimations [32], and the other had a median correlation of 97.9% (n = 20, range 94.2–99.3%) between fPCG and Doppler US recordings [45]. A further two studies which used fECG as a comparison reported a strong correlation (p<0.01, r = 0.96, n = 270) between beat-to-beat FHR signals [41], and a non-significant difference (p = 0.69, fPCG vs. fECG, 146.95 ± 8.16 BPM vs. 145.96 ± 8.15 BPM, n = 15) between the FHR values [37].

**Performance of combined fECG/fPCG devices.** Mhajna et al. [46] assessed the accuracy of the Invu system at recording the FHR using simultaneous CTG; a highly significant correlation was reported been the two modalities (r = 0.92, p<0.0001).

**Factors which affect performance of fECG.** Studies reported data on a variety of factors which could affect the performance of fECG devices: gestational age (15 studies), BMI (8 studies), time of day (5 studies), maternal movement (4 studies), fetal position (5 studies), location of recording (3 studies), maternal-fetal complications (5 studies), uterine activity (1 study), amniotic fluid index (1 study), multiple fetuses (1 study) and smoking status (1 study). These factors will be addressed in turn.

Fifteen studies investigated the association between the fetal gestational age at the time of the recording and device performance. The gestational age was broken down into distinct but unstandardized categories (e.g. $16^{+0}$ to $19^{+6}$ weeks' gestation) in ten of these studies [25,33,34,36,38,40,44,49,52,55], as shown in Fig 4. Overall, there was a reduction in the device performance in the middle gestational age categories, roughly from the start of the third trimester ($28^{+0}$ weeks) until 35–36 weeks' gestation. Moreover, the device performance was greatest at term or in post-term fetuses. Of the five studies not represented in Fig 4, a further two studies reported a decrease in the success between 24–34 weeks' gestation [58] and difficulty detecting a fECG signal at gestational ages 31 and 34 weeks [32]. The remaining studies

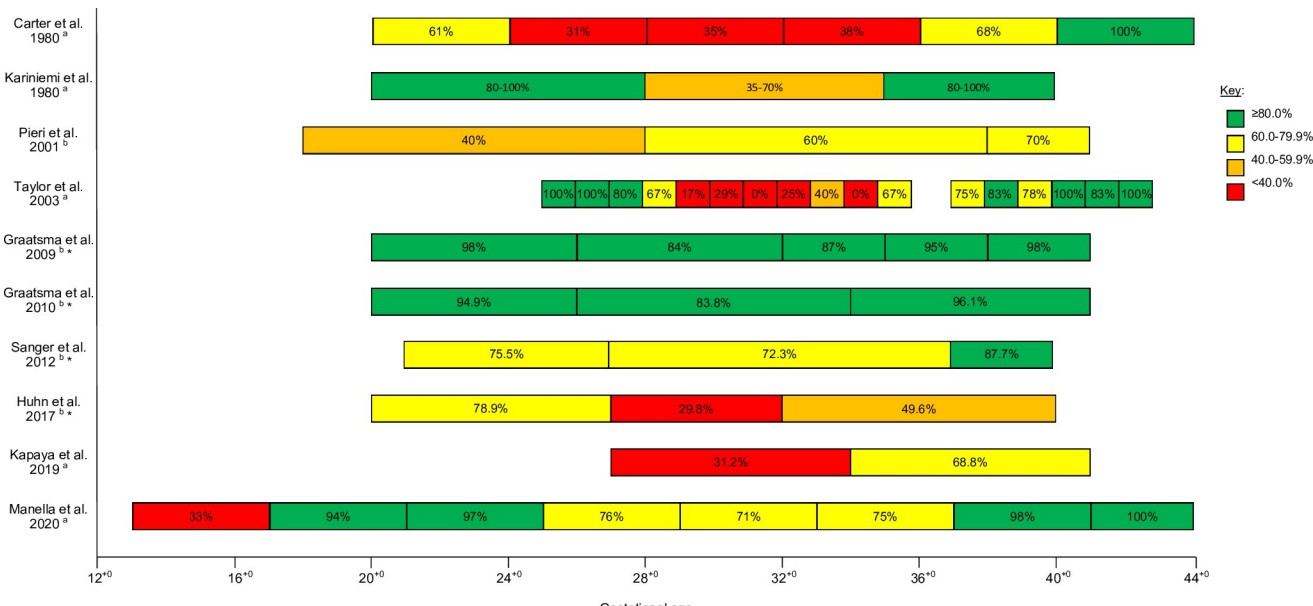

**Fig 4. Device performance of fetal electrocardiography devices at different fetal gestational ages (weeks).** The device performance (success rate or signal quality) for each study was extracted and reported in the figure using both the numerical value and a coloured key as a visual aid: green (≥80.0%), yellow (60.0–79.9%), orange (40.0–59.9%) and red (<40.0%). Four studies reported data only using a figure, hence data reported were extracted from the original figures either by drawing a line to the axis (Carter et al. and Graatsma et al. (2009)), or by dividing the number of successful recordings by the total number of recordings (Manella et al. and Taylor et al.). Kariniemi et al. did not provide a specific gestational age for "term"; for the purpose of this figure, term was assumed to be $40^{+0}$ weeks' gestation. Numerical values reported are as presented in the studies, or to the nearest whole number if extrapolated from a figure. [a] Values represent success rate (as reported by authors). [b] Values represent signal quality. * Studies which used the Monica AN24 device.

reported opposing findings: one found the majority of unsuccessful traces were between 37–39 weeks' gestation [30]; a single study found a weak positive correlation between the SQ and gestational age (p = 0.05) [26]; and one study determined that the relative gap duration between successful traces significantly decreased with increasing gestational age (p = 0.04) [56].

The effect of maternal BMI on the success rates was investigated in eight studies. One study found that the majority of participants whose fECG traces were unsuccessful had a BMI greater than 24.9 kg/m$^2$ [30], whereas another determined that BMI had no effect on the SQ apart from fetuses with a gestational age of 20 to $25^{+6}$ weeks, where BMI negatively correlated with the fECG SQ (p = 0.04) [34]. Furthermore, Van Leeuwen et al. [56] reported participants with higher BMIs had longer durations between valid FHR traces (p = 0.009) and a greater percentage of recording time with gaps (p = 0.03), as well as there being a trend to a lower proportion of valid fECG data being obtained (p<0.10). The remaining five studies reported that BMI had no significant effect on the device performance [26,33,36,52,54].

Three studies [26,38,51] provided comparable data investigating the association between time of day and SQ; SQ is significantly greater when recordings are taken at night in comparison to the overall SQ (Fig 5). A trend towards significance is also observed in the SQ of recordings taken 'at rest' or 'at night and at rest' compared to overall SQ. A further two studies [33,36] reported a greater SQ during the night.

When assessed, maternal movement always negatively affected the fECG device performance. Both Crawford et al. [26] and Huhn et al. [36] quantified the level of maternal movement using an arbitrary scale and reported significance levels of p<0.05 to p<0.0001. A further two studies reported the effect of signal loss due to maternal movements, although the fECG signals returned after a period of inactivity [27,29].

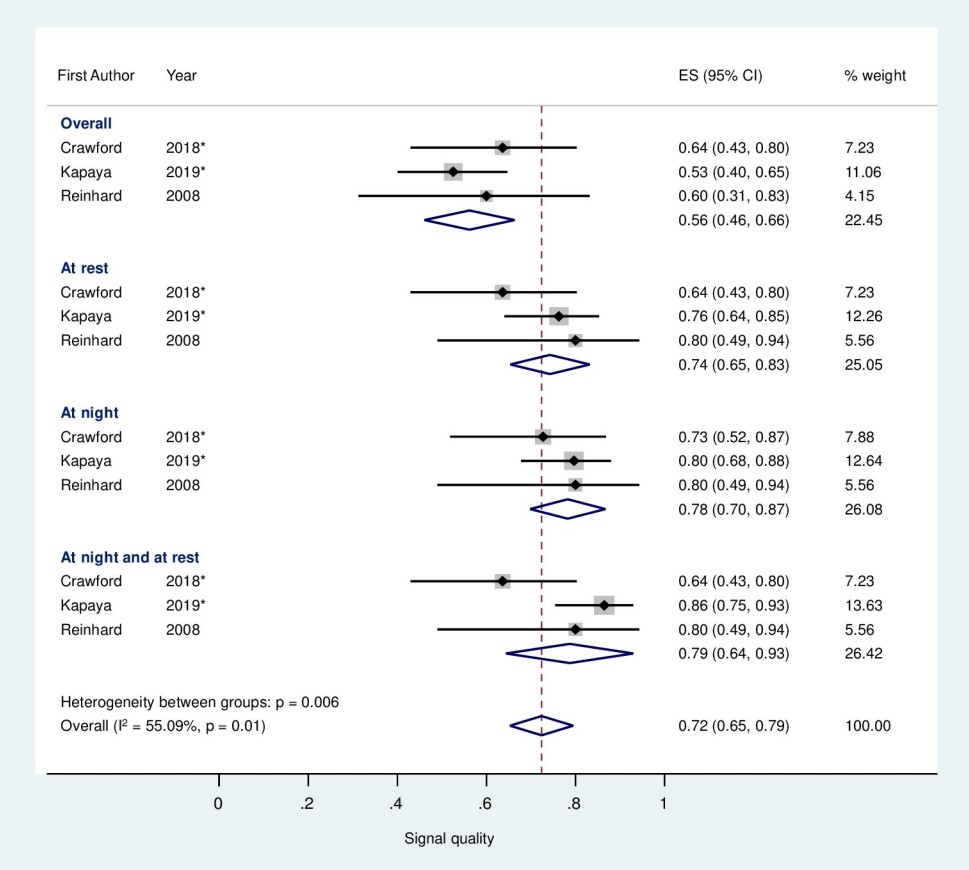

**Fig 5. Estimated average signal quality of fetal electrocardiograph recordings depending on time of day from three studies.** Black markers represent the signal quality with 95% confidence intervals (CI) (whiskers). The size of each grey square represents the relative weight in the meta-analysis. The diamond represents the signal quality summary value. Asterisk (*) represents studies which had the median and interquartile range/range data converted to the mean and standard deviation. All studies included used the Monica AN24 device.

The position of the fetus within the maternal abdomen was found to have no significant effect on the SQ in three studies [33,34,36], however the study conducted by Graatsma et al. [33] had a trend towards statistical significance (p = 0.06). Conversely, two studies stated that the fetal position had an effect on the fECG device [28,56]. One determined that fetuses in a breech or transverse position, when compared against those with cephalic presentation, had a greater proportion of gaps in their fECG traces (mean 65±30% vs. 29±23%; p = 0.008), a longer duration between valid FHR traces (2.2±0.5 seconds vs. 1.8±0.2 seconds; p = 0.02) and a trend towards a lower proportion of valid recording time (3±3% vs. 20±22%; p = 0.06) [56]. The other study simply stated "the quality of the fECG signals strongly depends on the position of the fetus inside the maternal abdomen" [28]; however this study only used a single participant and did not provide any quantitative data alongside this statement.

Two studies found no statistical difference in the SQ of recordings taken at home and those in the hospital [33,34]; however these studies simply compared the overall SQ against location. Following analysis of the SQ throughout 24 hours, a higher SQ was found in hospital group during the day (daytime: hospital 43.3% vs home 40.2%) but was lower in the night-time (night-time: hospital 71.1% vs home 86.8%), when compared to home SQ (p<0.001) [36].

An estimated fetal weight less than the 10<sup>th</sup> percentile was not reported to have any effect on the performance of fECG devices [36,39,56]; these three studies used different terminology for the small fetal size (fetal-growth restriction, small-for-gestational age and intrauterine growth restriction). Kapaya et al. [39] did report a large difference in the mean success rate of SGA (48.6%) fetuses compared to appropriate-gestational age fetuses (75.7%), however no statistical analysis was performed on this specific data to determine any significance. Graatsma et al. [34] stated that no maternal-fetal conditions affected the fECG device, however the authors did not fcne these conditions and this cohort of hospitalised women were compared to women who had home recordings, hence it is unclear whether this was a confounding variable. In addition, the success rate of fECG recordings was not affected by structural heart disease [58]. On the other hand, participants suffering from pre-eclampsia had a greater proportion of valid recording time (p = 0.01) and fewer gaps in the FHR trace per hour (p = 0.04) [56].

Only Crawford et al. [26] investigated the relationship of SQ and uterine activity and determined there was a strong negative correlation between these variables (p<0.001; $r^2$ = 0.79). One study [36] reported that women with a low amniotic fluid index ($\leq$5<sup>th</sup> percentile), also known as oligohydramnios, had a lower SQ compared to women with a normal index, although this difference was not significant (mean SQ: 12.0% vs. 48.5%; p = 0.096). This study did not report any effect of a high amniotic fluid index ($\geq$95<sup>th</sup> percentile) as no participants satisfied this criteria.

Taylor et al. [55] reported the fECG signal separation in singleton, twin and triplet pregnancies was successful in 85%, 78% and 93% of fetuses, respectively. All fetuses with separation success displayed clear P, Q, R and S waves, and T waves were able to be identified in 63%, 59% and 57% of successful traces, respectively. No trend, or lack of, between the number of fetuses and the success of fECG trace analysis was reported by the authors.

A single study [38] investigated the relationship between the maternal smoking status and the SQ: non-smokers had a significantly greater SQ in comparison to women who were current smokers (median SQ: 57.2% vs. 37.5%; p = 0.05). Overall, the included studies show that gestational age, maternal movement and the time of day have a clear effect on performance of fECG devices (Table 3). The effect of the remaining factors is unclear due to conflicting data and limited evidence from single studies in other cases.

## FM devices

Eight out of the 43 studies were specifically concerned with a FM device. Five of these studies utilised accelerometers, specifically the fetal movement acceleration measurement (FMAM) device [59,60,62,63] or the accelerometer-based fetal activity monitor (AFAM) device [66]. The remaining three studies utilised fVCG devices to quantify FM, all of which were unnamed [61,64,65].

**Table 3. Summary table showing the factors which appear to affect the performance of fetal electrocardiography devices and those which have a currently unclear effect.**

| Definite effect on device performance | Uncertain effect on device performance |
|---|---|
| • Gestational age<br>• Time of day<br>• Maternal movement | • BMI<br>• Fetal position<br>• Location of recording<br>• Uterine activity<br>• Amniotic fluid index<br>• Multiple fetuses<br>• Smoking status |

**Table 4. Fetal movement device descriptions.**

| Device name | Type of sensor/ electrode | Number of sensors/ electrodes | Arrangement of sensors/ electrodes | Placement of reference electrode | Signal processing | Additional information |
|---|---|---|---|---|---|---|
| *Accelerometers* | | | | | | |
| FMAM [59, 60, 62, 63] | Capacitive acceleration sensor. | 2 sensors. | 1 placed on the maternal abdominal wall to detect fetal movement (FM) and 1 placed on the mother's thigh to detect maternal movement (MM). | n/a | 500Hz sample rate with 5-30Hz BP filters. | The recorder contains 4 rechargeable batteries and 2 sensors, and weighs 290g. The sensor has 2 electrodes with capacitive acceleration, of which one is a movable diaphragm, and the other, a fixed back plate. Sensors attached to the abdomen using adhesive tape. |
| AFAM [66] | Analogue sensor. | 4 sensors. | 1 placed on maternal thorax, 2 placed on the lower abdomen, 1 placed close to the US probe (specific placement not mentioned). | n/a | Accelerometers sampled at 100Hz; root mean squared (RMS) of Euclidean norm of acceleration processed. | Sensors attached to the abdomen using adhesive tape. |
| fVCG devices | | | | | | |
| (*Unnamed Rooijakkers fVCG device*) [61] | *Not mentioned.* | 8 electrodes with 1 common reference electrode. | Placed in a circle around the umbilicus. | At the umbilicus. | Used a "NEMO" system. Filter 2Hz-98Hz BP (removes all out-of-band noise); notch at 50Hz (removes powerline interference); mECG detection followed by fECG while blanking all intervals ±50 ms around mECG R-peaks. | n/a |
| (*Unnamed Vullings (2008) fVCG device*) [64] | *Not mentioned* | 8 electrodes with 1 reference electrode and 1 common ground electrode. | Placed in a circle around the umbilicus. | At the umbilicus. | mECG dynamically segmented; mECG subsequently predicted and subtracted. fECG mean found from 30 complexes, with outliers rejected. fVCG determined from product of Moore-Penrose inverse of Dower matrix and sampled data. | Cables actively shielded. |
| (*Unnamed Vullings (2013) fVCG device*) [65] | *Not mentioned* | 8 electrodes | *Not mentioned.* | *Not mentioned.* | Signals acquired at 1 kHz, sampling rate using a NEMO system (Maastricht Instruments BV, the Netherlands), used a template subtraction method and Bayesian vectorcardiography method. | n/a |

AFAM: Accelerometer-based fetal activity monitor. BP: Band pass. fECG: Fetal electrocardiogram. FMAM: Fetal movement acceleration measurement. fVCG: Fetal vectorcardiography. mECG: Maternal electrocardiogram. MHR: Maternal heart rate. n/a: Not applicable. US: Usssltrasound.

**Technical description.** The technical details of the FM devices are described in Table 4. Two accelerometer device designs and detection technologies were presented, the FMAM device and AFAM. The devices differed in the number of sensors used, the former required two sensors and the later used four, as well as the arrangement of their sensors. Both devices required adhesive tape to attach the sensors to the maternal abdomen, however, the signal processing methods varied with the FMAM device having a sampling rate five times greater than the AFAM.

Three fVCG devices were presented, all of which appear to be from the same Dutch-based research group and hence required eight electrodes placed in a circle on the abdomen with a central reference electrode at the umbilicus, although one study [65] did not report the specific

placement of electrodes The signal processing methods greatly differed, however all required removal of the MHR trace.

**Device performance of accelerometers.** The four studies concerning the FMAM device described either the device success rate or the accuracy. Kamata et al. [59] and Ryo et al. (2018) [62] defined a successful recording as those with greater than four hours of recording time; the respective success rates being 94.3% and 75.3%. The remaining two studies [60,63] reported the accuracy using prevalence-adjusted bias-adjusted kappa (PABAK)—a measurement of the agreement between FM detected by the FMAM device and simultaneous US. Both reported similar mean PABAK values of 0.83 (SD±0.04) (n = 44) [60] and 0.79 (SD±0.12) (n = 45) [63] from 30 minute recordings. Mesbah et al. [66] reported the AFAM had an average accuracy of 55%, with a sensitivity and specificity of 59% and 54% respectively (n = 3).

**Performance of fVCG devices.** Two studies reported the sensitivity and specificity of fVCG devices against simultaneous US monitoring; one study [61] had a mean sensitivity of 67% (SD±24%) and a specificity of 90% (SD±8%) (n = 4) and the other [65] reported values of 47% and 87% respectively (n = 8). The remaining fVCG study [64] did not provide any quantitative information.

**Factors which affect performance of accelerometers.** Due to the limited number of studies including FM devices there is little data on the factors which affect the performance of these devices. Nonetheless, there is some degree of information regarding accelerometers, specifically the impact of gestational age (3 studies) and the positioning of the sensors on the maternal abdomen (1 study). Such information was not available for fVCG devices.

Two studies [60,63] which utilised the FMAM device reported increases in the accuracy (PABAK values) in late pregnancy; although neither study verified significance with statistical analysis. The study using the AFAM [66] showed a greater sensitivity achieved at 35 weeks' gestation in comparison to 32 weeks' gestation (76% vs. 50% and 52%, n = 3).

The position of the sensor on the maternal abdomen did not greatly alter the correlation (PABAK value) between gross FM detected using the FMAM device and US; a sensor positioned where the mother most strongly perceived FM had an overall correlation of 0.79 (SD ±0.12), in comparison to 0.76 (SD±0.15) on the opposite site across the abdominal midline [63].

## Discussion

This is the first review which studies the use of CFM devices in antenatal care, with specific interest in the performance of the devices and relevant factors which affect this, as well as the devices' design and the technologies employed to detect FHR or FMs. Fourty-three relevant articles were included which identified 24 different devices using four suitable technologies for CFM.

This review was strengthened by the use of a systematic search strategy using multiple databases and a broad scope which enabled the inclusion of a wide range of study designs, with no restrictions placed on the language or country of origin. However, any review is susceptible to publication bias and the omission of relevant articles, and quality assessments were not performed according to the design and conduct of a scoping review. In addition, it is important to acknowledge the fact that patented technologies which are not currently available in the public domain could exist, but for this reason could not be included. The lack of common device assessment in the source publications limited statistical analysis, and where meta-analysis was performed some non-normally distributed data was converted to approximated normally distributed values; the impact of this data alternation on the pooled estimates is unknown. Furthermore, the relative weights in the meta-analysis did not consider the replication of data in

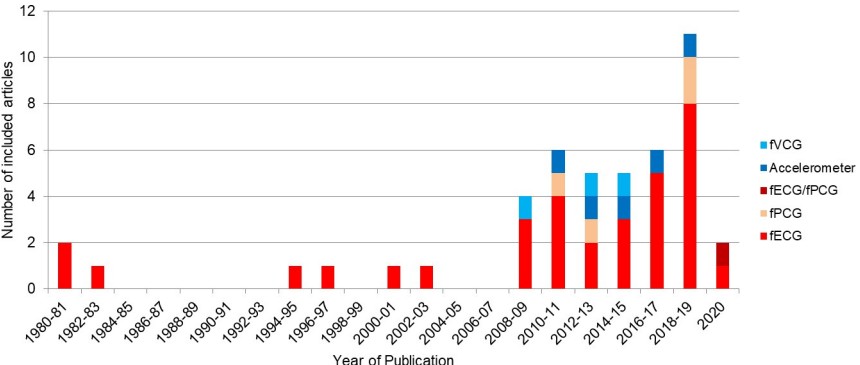

**Fig 6. The number of studies included in the review published every two years.** Devices have been split into the different technologies used: Fetal electrocardiography (fECG), fetal phonocardiography (fPCG), combined fECG/fPCG, accelerometry and fetal vectorcardiography (fVCG).

studies conducted by the same clinical research group as this information was not explicitly mentioned in any studies.

A variety of device designs and technological approaches were identified, however due to the reporting of data, we were unable to deduce a technology that appeared to be advantageous in terms of the reported device performance. However, fECG devices have been more widely investigated and hence offer a greater opportunity to be optimised and implemented into routine clinical practice than the other technologies. This may be because fECG devices are well established and the included fECG studies date back to 1980 [25], in comparison to more recent advances made in alternative devices such as fPCG (2008) [45], fVCG (2008) [64], accelerometers (2011) [66] and combined fECG/fPCG devices (2020) [46]. There has been increasing interest in CFM over the years; Fig 6 shows the studies included in this scoping review only. It is anticipated future CFM research will equally investigate both FHR and FM devices.

## Devices which monitor FHR

CFM devices which monitored the FHR were comprised of numerous fECG devices, fPCG devices and a single combined fECG/fPCG device. Currently only the Monica AN24 device, a fECG device, is in widespread use and was utilised in numerous studies, however its SQ exhibited high levels of variability between the different studies (mean SQ 68%, 95% CI 49–87%). Furthermore, studies with the largest cohort size showed the highest SQ suggesting that there is a possible learning curve in obtaining fECG recordings of optimal quality, however there was no statistically significant relationship between study size and SQ (p = 0.53, $r^2$ = 0.06). Whilst other FHR devices appear to have a better device performance, many have only been tested for short periods of time (median of 30 minutes) and hence their results must be viewed with caution with respect to CFM which would require longer-term monitoring. Various variables which could alter device performance were only evaluated in fECG device studies. A widely documented reduction in the performance of fECG devices was noticed between roughly 28 to 36 weeks' gestation. This has been attributed to electrical impedance from the vernix caseosa, a protective insulating layer which surrounds the fetus from 28 to 32 weeks' gestation and completely dissolves by 37 weeks' gestation [67]. The vernix caseosa has an impedance factor of 500–1000 higher than amniotic fluid [68,69] and hence significantly reduces the fECG amplitude, however the FHR can still be weakly detected due to signal transmission through the umbilical cord, oronasal cavity and gaps within the vernix caseosa [67,69]. Therefore, as the impact of gestational age on fECG trace quality is widely understood,

further studies are not required to test this specifically but gestational age should be taken into consideration when evaluating the impact of other factors. Regarding novel fECG devices, ideally these should be initially tested after 37 weeks' gestation to minimise signal disruption, as this will elucidate whether the devices can accurately and reliably extract FHR and/or FM data.

In the majority of studies, BMI had no effect on the signal quality of fECG devices, hence fECG appears advantageous to conventional US-based technologies which are negatively affected by increasing maternal BMI [70]. Critically some studies deviated from this finding stating an opposing effect.

Maternal movement is a clear limitation of current CFM technology. Abdominal wall muscle contractions cause fECG noise interference with frequencies up to 500 Hz, often preventing reliable detection of FHR traces [36,71]. This further clarifies the observed decrease in daytime success rates, compared to night-time, due to high levels of maternal movement in a woman's day-to-day lives. Moreover, although the location (hospital vs. home) of fECG recordings does not affect the overall success rates, day-time recordings are more successful in the hospital setting as women will move less and are often confined to their bed, whereas in the night-time recordings are less successful in hospital than at home as women could have a lower quality of sleep due to the novel environment, ambient noise and interruptions by health care professionals and other patients [36,72].

Antenatal fECG devices have been widely extrapolated from those developed for use in intrapartum care, thus in theory the devices should be able to provide a sufficient SQ regardless of the level of uterine activity. Nonetheless, Crawford et al. [26] reported a strong negative correlation between uterine activity and SQ ($p<0.001$; $r^2 = 0.79$). One explanation could be that that uterine activity may mimic the effect of maternal physical activity causing disruption of the FHR signals. However, uterine contractions can trigger FHR accelerations [73], a reassuring sign of fetal well-being. It is important that CFM FHR devices have the ability to detect these significant changes in the FHR variability, as lack of accelerations indicate fetal compromise requiring possible clinical intervention. Whilst other factors may or may not affect the quality and success of fECG devices, further research is required to deduce those which have a significant impact.

All FHR devices in the presented studies had different designs and technological approaches. The studies failed to provide a clear rationale for the quantity of electrodes or sensors and the specific arrangement used. With respect to fECG devices, the electrode configuration can significantly influence the FHR signal quality; the number of electrodes must be optimised to maximise the signal-to-noise ratio whilst minimising power consumption, as well as considering the electrode orientation and placement [71]. A Dutch research group have proposed the use of five electrodes is optimal in the third trimester, whilst an extra electrode is required earlier in pregnancy due to a variable fetal position [71].

Whilst fECG devices have been the subject of a large proportion of studies, this technology does not appear to be advantageous to fPCG, which is relatively understudied. Mhanja et al. [46] successfully designed and tested a device which incorporated both fECG and fPCG technologies, the Invu system. Although this device successfully demonstrated the ability to combine two FHR technologies and showed a highly significant correlation with CTG ($r = 0.92$, $p<0.0001$), further evaluation is required with larger sample sizes for longer durations to determine feasible clinical utility.

## Devices which monitor FM

FM devices comprised of fVCG technology and accelerometers, however the device performance was not comparable between the two technologies. In addition, the factors which affect the device performance were only assessed in accelerometers.

The correlation of FM detected by accelerometers compared to simultaneous US were greater in late pregnancy. This relationship has been attributed to the increasing strength of FM causing greater abdominal wall oscillations, which occurs throughout the progression of pregnancy [63]. On the other hand, a recent study conducted by Verbruggen et al. [74] reported the fetal kick force increases throughout pregnancy until 30 weeks' gestation and subsequently reduces until birth due to mechanical stress and strain. The included studies suggest FM devices may be more effective towards the end of pregnancy, although this was observed in a small cohort (n = 92) with a limited gestational age categories. Future use of larger cohorts will provide a clear overview of the association (or lack of) between the performance of accelerometers and fetal gestational age. Although not investigated using accelerometers, concerns have been raised about the impact of respiratory artefacts caused by sleep apnoea, a common co-morbidity of obesity, on the quality of accelerometer recordings [60].

The studies concerning the FMAM device [59,60,62,63] provided a clear rationale for the use of two sensors and their specific placement on the maternal abdomen and thigh. The other accelerometer device, the AFAM, and the fVCG devices did not justify the specific device design. Currently, the only FM device which has been studied for prolonged periods of time is the FMAM, which was used overnight in three studies [59,62,63]. Therefore the FMAM appears to be more technologically advanced to have the ability to record FM overnight, compared to the remainder of FM devices which have only been studied for a maximum of 30 minutes. Substantial changes in the FM pattern can be indicative of fetal distress and often act as a 'warning sign' prior to stillbirth [1,2]. However, FM patterns vary significantly between individuals and can alter weekly as pregnancy progresses [60]. At present, this demonstrates a key drawback of FM devices, and that considerable research is required to develop reliable and widely applicable FM count indexes to ensure CFM devices can accurately detect fetuses whose FM pattern deviates from normality.

## Further advancements required in CFM

With increasing interest in CFM devices it would be beneficial to develop a standardised and systematic format of device assessment and reporting to aid comparison between the various devices. One proposition would be to determine the sensitivity, specificity and accuracy against concurrent use current gold standard methods using short recording periods (e.g. 20–30 minutes), and to report the signal quality in longer (i.e. >90 minutes) recordings. This approach would be applicable to both FHR and FM devices and will determine the suitability of devices for CFM, ensuring the FHR or FM pattern can be reliably recorded for long periods of time. It is anticipated that a standardised method of reporting will highlight favourable approaches. For CFM devices to be implemented into clinical practise, additional issues must first be addressed. Whether the device is primarily analysing the FHR or FM pattern, analysis must be individualised to each fetus and account for changes which occur in these parameters as pregnancy progresses [60], however theoretically 'normal' patterns must also be known. This includes the normal fetal sleep patterns, response to uterine contractions and the normal fetal movement pattern throughout the day. This awareness will ensure any deviation from the normal FHR or FM pattern is detected via the CFM device.

A significant reduction or sudden alteration in FM acts as a 'warning' sign prior to fetal death [75]; this pathological change is detected in 31–55% of cases by maternal instinct in the preceding week of stillbirth [75–77]. Therefore, another potential important development could be adding an interactive component for mothers to report significant events via a mobile phone application to aid clinical analysis and provide maternal reassurance that their possible

concerns are being monitored. This could include detailing periods of gross fetal movement or lack of, instances of uterine contractions and other symptoms such as abdominal pain.

The current use of intermittent monitoring throughout high-risk pregnancies does undoubtedly provide reassurance to pregnant women, and helps to relieve anxiety. The implementation of CFM devices into clinical practice could reduce or potentially replace the need for current methods. However, concerns have been raised by women about the sole use of CFM in antenatal care, and many currently perceive CFM as an 'add-on' form of monitoring to current methods [26]. Nonetheless, the experimental use of CFM devices has already prevented adverse outcomes in a number of cases, specifically using the Monica AN24 [50] and the FMAM [78]. In addition to this, as well as providing reassurance to women that their baby is being actively monitored [17], the use of a CFM device increases maternal awareness of fetal wellbeing [60]. This demonstrates the multiple benefits which can be achieved through the use of CFM in clinical practice, enabling timely identification of compromised fetuses which in turn could assist in the reduction of the stillbirth and neonatal mortality rates. Nonetheless, whilst clinical studies are still undergoing to assess the reliability of CFM it is also important that intermittent monitoring continues to take place so there is no deviation from current 'gold-standard' forms of practice.

## Conclusions

In conclusion, CFM could alleviate the intermittent nature of current antenatal fetal monitoring methods, providing an objective and longitudinal overview of fetal wellbeing. To date, numerous different CFM devices have been developed to address this need; however there is a high level of inter-device and intra-device variability and currently no approach appears to be advantageous. In addition, there appear to be numerous factors which affect the quality of CFM recordings, although these have only been investigated in fECG monitors and accelerometers. It is clear that gestational age, maternal movement and the time of day clearly alter device performance, however the evidence base for other factors such as the impact of BMI, uterine activity and the amniotic fluid index is sparse. Consequently, additional studies are required to specifically highlight the impact of such factors as this will help to aid the development of better devices and highlight certain pregnancies where the device's quality and diagnostic ability is reduced.

Overall, although CFM appears to be a viable form of fetal monitoring, at present the utility of CFM devices in routine clinical care cannot be strongly recommended due to the wide disparities between studies alongside the unclear impact of certain maternal and fetal factors. In order for this recommendation to be reviewed, first the devices must have reduced device performance variability and undergo further rigorous testing to ensure they can detect alterations in the FHR and/or the FM pattern, enabling prompt detection of fetal compromise.

## Supporting information

**S1 Table. Literature search history.** A detailed description of the electronic literature searches.
(DOCX)

**S2 Table. Articles excluded after full text assessment.** A list of the 63 articles that were excluded after full-text assessment and the reason for exclusion.
(DOCX)

**S1 File. PRISMA 2009 checklist.**
(DOC)

## Acknowledgments

The authors wish to thank Elham Aalia from the Trust Library Services at Manchester University NHS Foundation Trust for providing training at using the relevant databases and for assistance in developing the systematic search strategy.

## Author Contributions

**Conceptualization:** Kajal K. Tamber.

**Data curation:** Kajal K. Tamber, Stephen J. Carey, Jayawan H. B. Wijekoon, Alexander E. P. Heazell.

**Formal analysis:** Kajal K. Tamber, Dexter J. L. Hayes, Alexander E. P. Heazell.

**Funding acquisition:** Jayawan H. B. Wijekoon, Alexander E. P. Heazell.

**Investigation:** Kajal K. Tamber.

**Methodology:** Kajal K. Tamber, Dexter J. L. Hayes, Alexander E. P. Heazell.

**Project administration:** Kajal K. Tamber, Alexander E. P. Heazell.

**Resources:** Alexander E. P. Heazell.

**Supervision:** Alexander E. P. Heazell.

**Validation:** Alexander E. P. Heazell.

**Visualization:** Kajal K. Tamber, Dexter J. L. Hayes, Stephen J. Carey, Jayawan H. B. Wijekoon, Alexander E. P. Heazell.

**Writing – original draft:** Kajal K. Tamber, Dexter J. L. Hayes, Stephen J. Carey, Jayawan H. B. Wijekoon, Alexander E. P. Heazell.

**Writing – review & editing:** Kajal K. Tamber, Dexter J. L. Hayes, Stephen J. Carey, Jayawan H. B. Wijekoon, Alexander E. P. Heazell.

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
