## [Decision Letter · Decision Letter 0]

20 Aug 2020

PONE-D-20-20354

A systematic scoping review to identify the design and assess the performance of devices for antenatal continuous fetal monitoring

PLOS ONE

Dear Dr. Tamber,

Thank you for submitting your manuscript to PLOS ONE. After careful consideration, we feel that it has merit but does not fully meet PLOS ONE’s publication criteria as it currently stands. Therefore, we invite you to submit a revised version of the manuscript that addresses the points raised during the review process.

The manuscript has been reviewed by three leading experts in the field. I invite you to address all their comments point by point. 

We look forward to receiving your revised manuscript.

Kind regards,

Martin G Frasch, MD, PhD

Academic Editor

PLOS ONE

Journal Requirements:

2. Please ensure you have thoroughly discussed any potential limitations of this study within the Discussion section.

Reviewers' comments:

Reviewer's Responses to Questions

**Comments to the Author**

1. Is the manuscript technically sound, and do the data support the conclusions?

Reviewer #1: Yes

Reviewer #2: Yes

Reviewer #3: Yes

2. Has the statistical analysis been performed appropriately and rigorously? 

Reviewer #1: Yes

Reviewer #2: Yes

Reviewer #3: Yes

3. Have the authors made all data underlying the findings in their manuscript fully available?

Reviewer #1: Yes

Reviewer #2: Yes

Reviewer #3: Yes

4. Is the manuscript presented in an intelligible fashion and written in standard English?

Reviewer #1: Yes

Reviewer #2: Yes

Reviewer #3: Yes

5. Review Comments to the Author

Reviewer #1: The authors present a review paper on devices for antenatal continuous fetal monitoring. Such a paper would be interesting to the field to see how far this relevant technology has evolved and what work still needs to be done before we can except wide embracing of this technology in the clinics.

Although the authors have read an impressive amount of papers, I think their results are still incomplete. Relevant works/devices that seem to be missing:

Phonocardiography: Kovacs et al. (DOI: 10.1109/TBME.2010.2071871)

Fetal electrocardiography: Cohen et al. (DOI: 10.1111/j.1600-0412.2012.01533.x). This paper discusses a device durante partu, but as the authors claim themselves in the discussion on line 444 that these devices should be evaluated after 37 weeks’ gestation, I think it would be appropriate to include this work, especially as it present a large quantitative evaluation.

Fetal movement/fetal VCG: Vullings et al. (doi: 10.1109/TBME.2013.2238938) which includes more patients and analysis than the proceedings used in the review.

Devices:

Mindchild Meridian ( https://www.ajog.org/article/S0002-9378(09)01641-X/pdf)

Nemo Healthcare (doi: 10.1111/aogs.13873. and doi: 10.1016/j.earlhumdev.2019.01.011)

Nuvo (https://www.ahajournals.org/doi/abs/10.1161/circ.140.suppl_1.13151)

At the moment, I think it is not appropriate to provide detailed comments on the full document because I would expect the inclusion of the works above to lead to rather large modifications.

Yet, I think the work is already so extensive and relevant that the authors should be granted the opportunity to extend their work and have it published.

A few things that I noticed while reading:

Table 2: Anderotti should be Andreotti

Line 373: vectrocardiography should be vectorcardiography

Reviewer #2: General

The authors provide an important overview of the rapidly changing landscape of fetal monitoring. The relevance of this effort is made evident and timely following the impact of the recent COVID-19 pandemic and the gradual transition to remote healthcare. The methodological systematic approach of the review utilizing multiple databases and the description of the results is apparent and should be commended. However, the role of this review paper in the literature can and should be expanded beyond a description of the current literature (see below). Briefly, the clinical value of continuous fetal monitoring (CFM) is not made clear, and as such, the different clinical aspects of it and their separate success criteria are not well defined. Furthermore, underlying technologies of the different clinical tools utilized during CFM, such as FHR, fECG, and FM, are all pooled and compared together. Better framing of the underlying hypothesis and the research questions that this review raises and answers can help clarify the manuscript.

Intro

The notion of continuous fetal monitoring is introduced in the introduction (line 83) – it is not clear what is defined as continuous fetal monitoring. What differs continuous fetal monitoring from standard fetal monitoring performed during the antepartum stage? In this aspect, it would be helpful to define the term ‘prolonged periods of time’, used in different locations in the manuscript. Furthermore, it is implied that longer monitoring times may lead to or provide better clinical outcomes. What is the supporting evidence of this hypothesis? What is the suggested mechanism of action? Past work by the authors has shown preference of women’s experience towards continuous monitoring [1] but did not provide clinical evidence that this method is indeed preferable.

In addition, the results describe three different methods of continuous monitoring - capturing fetal heart rate, capturing fetal ECG, and capturing fetal movement over ‘prolonged periods of time’. It is not described why these three specific methods were selected, and again, what is the clinical value of each one of them when collected over time. It is reasonable to assume that each one of these methods provides very different clinical insights.

As the clear clinical value of continuous fetal monitoring is not described, it is not made clear as to what element or elements of continuous fetal monitoring is most important - is it the accuracy of the device? Is it the ability to capture the fetal heart rate signal over time? Is it the ability to reduce anxiety of the mother, or perhaps it is the ability to provide the physician with more information or the ability to wear the device continuously by the patient? Similarly, one may ask why should fetal movement and fetal ECG be included in this review? Has continuous FM monitoring clinically shown to improve outcome? In lack of existing evidence, it is suggested to provide a hypothesis as to why continuous fetal monitoring should improve outcome.

Without a clear set of success criteria, the attempt to ‘...provide a systematic overview of different CFM devices’ and specifically ‘...map the devices’ design, performance and factors which affect this, whilst determining any gaps in development’ is challenging. Clearly, due to the nature of the review paper, one is limited to the information that exists in the different studies. However, attempting to define the success criteria may be one of the main contributions of this kind of review paper (see remarks on discussion).

Methods

The search strategy, the study screening and selection process, data extraction, data presentation and data synthesis all seem intact. Notedly, a similar approach was successfully performed in past work by the authors [1].

Results

The results reported regarding the different factors which affect performance, such as BMI, gestational week, time of day, etc, are interesting and relevant for this kind of review.

A recent publication [2] describing a detailed validation study of a new device for antepartum continuous remote fetal monitoring which seems to meet all criteria to be included in this study is currently lacking in the review. It is recommended to add this study to the review.

Another recent paper [3] is a high quality study comparing the Monica Novii system to CTG. It seems that this paper should also be part of the review as well.

A result that would be interesting to see in this kind of review is whether there is a growing trend in the literature of continuous fetal monitoring (e.g. the total amount of research over the years, year by year). This kind of analysis may provide some insight on the clinical approach towards CFM.

Discussion:

The discussion raises the issue of not being able to deduce an advantageous technology (line 420). However, it is not clear for what purpose these technologies should be advantageous for. There seems to be a mix up between the underlying technology (fECG, accelerometers, etc) and the clinical utilization of the technology (FHR, FM, etc). The question of advantageous technology should be asked for each clinical utility separately.

Following the points raised related to the introduction, a review of this type can be a good opportunity to try and define a framework for success criteria for CTM devices. The reviewers can try to suggest what are the important elements of continuous monitoring and as a result what may be plausible success/fail criteria for such devices. This kind of a discussion can help advance the field of fetal monitoring. The conclusion section does raise this point, but it can be further expanded in the discussion.

References:

[1] Crawford A, Hayes D, Johnstone ED, Heazell AEP. Women’s experiences of continuous fetal monitoring – a mixed-methods systematic review. Acta Obstet Gynecol Scand 2017; 96:1404–1413.

[2] Mhajna M, Schwartz N, Levit-Rosen L, et al. Wireless, remote solution for home fetal and maternal heart rate monitoring. Am J Obstet Gynecol: MFM 2020

[3] Monson M et al. Evaluation of an external fetal electrocardiogram monitoring system: a randomized controlled trial. Am J Obstet Gynecol. 2020; 223:244.e1-244.e12

Reviewer #3: The efforts by the authors are to be commended. Their search of the literature to provide a systematic overview of devices for continuous antenatal surveillance was exhaustive. They achieved their objectives of mapping the design, and other factors affecting performance,of the various available devices, while determining any gaps in development. In this they have succeeded admirably. The evaluation and the determination of the clinical utility of these devices, however, was hampered by the absence of common means of evaluating device performance or comparing various devices for these purposes.

Thus, the issues of clinical utility, practical acceptance on a broad scale, and the physiological underpinning of the devices remain in question, even assuming their functioning as advertised.

The benefits of fetal monitoring during labor relate to the assessment of the responses of the fetal heart rate pattern to the recurrent provocation of stressful uterine contractions. With modest hypoxia, the fetus responds to contractions immediately with late decelerations. It may be expected, that with absent or infrequent contractions, changes in the fetal heartrate pattern will develop more slowly. Under these circumstances, behavior will become affected and over time, the baseline rate will begin to rise and variability will become reduced.

Continuous monitoring in the antepartum period, therefore, must utilize evaluations that are, in part, different from those used in labor with greater attention paid to the detection of normal fetal behavior and the regularity and response to fetal activity as well as the occasional contraction (Braxton Hicks).

Whether the methodology rests with FHR or fetal movement, the analysis must include the notion of fetal behavior, that is, the normal rhythmic fetal behavior (sleep patterns), and responses to contractions and fetal movements.

In this vein, the authors might vouchsafe that consideration be given by the manufacturers of these devices to providing information into the system that reflects the mother’s perception of fetal activity and uterine contractions to provide a marker against which any changes in fetal heart rate or fetal pattern of activity can be assessed.

6. PLOS authors have the option to publish the peer review history of their article (what does this mean?). If published, this will include your full peer review and any attached files.

Reviewer #1: No

Reviewer #2: No

Reviewer #3: No

---

## [Author Response · Author response to Decision Letter 0]

3 Oct 2020

The authors would like to thank the editor and reviewers for their comments on our manuscript, particularly during a time when clinicians and academics have been very busy adapting to the challenges posed by COVID-19. We have addressed each of the comments in detail below and amended our manuscript accordingly. The changes and responses are shown in blue font. 

"Reviewer #1: 

The authors present a review paper on devices for antenatal continuous fetal monitoring. Such a paper would be interesting to the field to see how far this relevant technology has evolved and what work still needs to be done before we can except wide embracing of this technology in the clinics.

Although the authors have read an impressive amount of papers, I think their results are still incomplete. Relevant works/devices that seem to be missing:

• Phonocardiography: Kovacs et al. (DOI: 10.1109/TBME.2010.2071871)

• Fetal electrocardiography: Cohen et al. (DOI: 10.1111/j.1600-0412.2012.01533.x). This paper discusses a device durante partu, but as the authors claim themselves in the discussion on line 444 that these devices should be evaluated after 37 weeks’ gestation, I think it would be appropriate to include this work, especially as it present a large quantitative evaluation.

• Fetal movement/fetal VCG: Vullings et al. (doi: 10.1109/TBME.2013.2238938) which includes more patients and analysis than the proceedings used in the review."

Thank you for highlighting the above papers. As indexing and search terms for observational studies are not as well developed as for intervention studies it is inevitable that a search strategy will not have 100% sensitivity (i.e. will miss papers). We’re very grateful to the reviewer for highlighting this and we have now included the first (Kovács) and third (Vullings) studies in the review. We have not included the second study (Cohen) as this study investigated intrapartum monitoring, which is part of our strict exclusion criteria. We agree that this would have been a useful study to include about the accuracy of the Monica AN24 monitor, but feel it is important to maintain focus on antepartum monitoring. 

"Devices:

• Mindchild Meridian (https://www.ajog.org/article/S0002-9378(09)01641-X/pdf)

• Nemo Healthcare (doi: 10.1111/aogs.13873. and doi: 10.1016/j.earlhumdev.2019.01.011)

• Nuvo (https://www.ahajournals.org/doi/abs/10.1161/circ.140.suppl_1.13151)"

The authors were aware of the Mindchild Meridian device before conducting the scoping review, and within the current literature it has only been utilised in intrapartum care. Due to our eligibility criteria, these papers (and hence this device) were excluded from the scoping review, which focuses only on devices which have been tested during the antepartum period of pregnancy. In addition the first Nemo Healthcare paper (doi: 10.1111/aogs.13873) only assessed the device in intrapartum care, hence we are unable to include this. Unfortunately the ‘Nuvo’ device was only referenced as part of a conference abstract – these were excluded from the scoping review due to the inability to obtain a relevant amount of data. 

The second Nemo Health care paper (doi: 10.1016/j.earlhumdev.2019.01.011) was missed in our initial searches, and it has now been included in the review as it meets all of the inclusion criteria. Thank you for highlighting this paper to us. 

At the moment, I think it is not appropriate to provide detailed comments on the full document because I would expect the inclusion of the works above to lead to rather large modifications.

Yet, I think the work is already so extensive and relevant that the authors should be granted the opportunity to extend their work and have it published.

A few things that I noticed while reading:

Table 2: Anderotti should be Andreotti

Line 373: vectrocardiography should be vectorcardiography"

We have amended these typographical errors in the re-submitted manuscript.

"Reviewer #2: 

General

The authors provide an important overview of the rapidly changing landscape of fetal monitoring. The relevance of this effort is made evident and timely following the impact of the recent COVID-19 pandemic and the gradual transition to remote healthcare. The methodological systematic approach of the review utilizing multiple databases and the description of the results is apparent and should be commended. However, the role of this review paper in the literature can and should be expanded beyond a description of the current literature (see below). Briefly, the clinical value of continuous fetal monitoring (CFM) is not made clear, and as such, the different clinical aspects of it and their separate success criteria are not well defined. Furthermore, underlying technologies of the different clinical tools utilized during CFM, such as FHR, fECG, and FM, are all pooled and compared together. Better framing of the underlying hypothesis and the research questions that this review raises and answers can help clarify the manuscript.

Intro

The notion of continuous fetal monitoring is introduced in the introduction (line 83) – it is not clear what is defined as continuous fetal monitoring. What differs continuous fetal monitoring from standard fetal monitoring performed during the antepartum stage? In this aspect, it would be helpful to define the term ‘prolonged periods of time’, used in different locations in the manuscript. "

The authors have defined continuous fetal monitoring in the introduction section and in the methods. Current methods of antepartum fetal monitoring (i.e. antepartum cardiotocography / non-stress test) are rarely used for longer than 90 minutes. We have added this time period to the manuscript. 

"Furthermore, it is implied that longer monitoring times may lead to or provide better clinical outcomes. What is the supporting evidence of this hypothesis? What is the suggested mechanism of action? Past work by the authors has shown preference of women’s experience towards continuous monitoring [1] but did not provide clinical evidence that this method is indeed preferable."

As the reviewer notes this is a hypothetical benefit at the present time. Importantly, there is no evidence that current methods used in clinical practice (i.e. antepartum cardiotocography / non-stress test) improve perinatal outcomes for the infant (see reference 12). The authors are not aware of any studies that have compared the effectiveness of continuous vs. intermittent fetal monitoring. We have clarified the introduction to ensure readers are aware this is a hypothetical benefit. 

"In addition, the results describe three different methods of continuous monitoring - capturing fetal heart rate, capturing fetal ECG, and capturing fetal movement over ‘prolonged periods of time’. It is not described why these three specific methods were selected, and again, what is the clinical value of each one of them when collected over time. It is reasonable to assume that each one of these methods provides very different clinical insights."

The authors describe these methods because these are the approaches that were identified using the search strategy. We have outlined the pathway of fetal compromise and described the points at which these approaches may detect changes in fetal physiology / behaviour. 

"As the clear clinical value of continuous fetal monitoring is not described, it is not made clear as to what element or elements of continuous fetal monitoring is most important - is it the accuracy of the device? Is it the ability to capture the fetal heart rate signal over time? Is it the ability to reduce anxiety of the mother, or perhaps it is the ability to provide the physician with more information or the ability to wear the device continuously by the patient? Similarly, one may ask why should fetal movement and fetal ECG be included in this review? Has continuous FM monitoring clinically shown to improve outcome? In lack of existing evidence, it is suggested to provide a hypothesis as to why continuous fetal monitoring should improve outcome."

The authors respectfully draw the reviewer’s attention to the purpose of the manuscript which was 1) to describe the design and detection technology employed in the devices; 2) compare the device performance of the different continuous fetal monitoring devices; and 3) investigate factors which affect the devices’ performance. It is evident that before the efficacy of any device can be evaluated in pregnant women the device must be accurate and reliable. Therefore, we included all technologies that monitored a quantifiable aspect of fetal wellbeing as we do not know which (if any) of these approaches might be more effective in continuous fetal monitoring. We have added a section of text to the introduction to summarize the hypothesis why continuous fetal monitoring might improve outcome.

"Without a clear set of success criteria, the attempt to ‘...provide a systematic overview of different CFM devices’ and specifically ‘...map the devices’ design, performance and factors which affect this, whilst determining any gaps in development’ is challenging. Clearly, due to the nature of the review paper, one is limited to the information that exists in the different studies. However, attempting to define the success criteria may be one of the main contributions of this kind of review paper (see remarks on discussion)."

As this is a scoping review its chief aim is to describe the field of published work. The authors agree that describing the variation between studies is an important outcome of our review and have addressed this in the discussion.

"Methods

The search strategy, the study screening and selection process, data extraction, data presentation and data synthesis all seem intact. Notedly, a similar approach was successfully performed in past work by the authors [1].

Results

The results reported regarding the different factors which affect performance, such as BMI, gestational week, time of day, etc, are interesting and relevant for this kind of review.

A recent publication [2] describing a detailed validation study of a new device for antepartum continuous remote fetal monitoring which seems to meet all criteria to be included in this study is currently lacking in the review. It is recommended to add this study to the review."

Thank you for highlighting the above paper to us, the authors have ensured this has been included in the revised manuscript. As our searches were performed in February 2020 we believe this is the reason the paper was missed, as it was published in May 2020. 

"Another recent paper [3] is a high quality study comparing the Monica Novii system to CTG. It seems that this paper should also be part of the review as well."

The authors did not include the Monica Novii system due to our eligibility criteria which does not include studies which are conducted during labour (as monitoring in this context is conducted differently). In addition, this device has been developed for use in intrapartum care only and therefore is not suitable as a monitoring device in antenatal care. 

"A result that would be interesting to see in this kind of review is whether there is a growing trend in the literature of continuous fetal monitoring (e.g. the total amount of research over the years, year by year). This kind of analysis may provide some insight on the clinical approach towards CFM."

Thank you for this suggestion – we believe this information would be extremely useful to show the development of the field and and have included it in Figure 6 as part of the discussion. 

"Discussion:

The discussion raises the issue of not being able to deduce an advantageous technology (line 420). However, it is not clear for what purpose these technologies should be advantageous for. There seems to be a mix up between the underlying technology (fECG, accelerometers, etc) and the clinical utilization of the technology (FHR, FM, etc). The question of advantageous technology should be asked for each clinical utility separately." 

The authors have amended the discussion to separate the technologies and their utilization in order that the question of advantageous technology can be addressed for the different clinical utility.

"Following the points raised related to the introduction, a review of this type can be a good opportunity to try and define a framework for success criteria for CTM devices. The reviewers can try to suggest what are the important elements of continuous monitoring and as a result what may be plausible success/fail criteria for such devices. This kind of a discussion can help advance the field of fetal monitoring. The conclusion section does raise this point, but it can be further expanded in the discussion."

This point has been developed further in the discussion. We have made some recommendations for elements that should be reported when evaluating devices for continuous fetal monitoring and criteria for success.

"References:

[1] Crawford A, Hayes D, Johnstone ED, Heazell AEP. Women’s experiences of continuous fetal monitoring – a mixed-methods systematic review. Acta Obstet Gynecol Scand 2017; 96:1404–1413.

[2] Mhajna M, Schwartz N, Levit-Rosen L, et al. Wireless, remote solution for home fetal and maternal heart rate monitoring. Am J Obstet Gynecol: MFM 2020

[3] Monson M et al. Evaluation of an external fetal electrocardiogram monitoring system: a randomized controlled trial. Am J Obstet Gynecol. 2020; 223:244.e1-244.e12

Reviewer #3: 

The efforts by the authors are to be commended. Their search of the literature to provide a systematic overview of devices for continuous antenatal surveillance was exhaustive. They achieved their objectives of mapping the design, and other factors affecting performance, of the various available devices, while determining any gaps in development. In this they have succeeded admirably. The evaluation and the determination of the clinical utility of these devices, however, was hampered by the absence of common means of evaluating device performance or comparing various devices for these purposes.

Thus, the issues of clinical utility, practical acceptance on a broad scale, and the physiological underpinning of the devices remain in question, even assuming their functioning as advertised.

The benefits of fetal monitoring during labour relate to the assessment of the responses of the fetal heart rate pattern to the recurrent provocation of stressful uterine contractions. With modest hypoxia, the fetus responds to contractions immediately with late decelerations. It may be expected, that with absent or infrequent contractions, changes in the fetal heartrate pattern will develop more slowly. Under these circumstances, behaviour will become affected and over time, the baseline rate will begin to rise and variability will become reduced.

Continuous monitoring in the antepartum period, therefore, must utilize evaluations that are, in part, different from those used in labour with greater attention paid to the detection of normal fetal behaviour and the regularity and response to fetal activity as well as the occasional contraction (Braxton Hicks).

Whether the methodology rests with FHR or fetal movement, the analysis must include the notion of fetal behaviour, that is, the normal rhythmic fetal behaviour (sleep patterns), and responses to contractions and fetal movements.

In this vein, the authors might vouchsafe that consideration be given by the manufacturers of these devices to providing information into the system that reflects the mother’s perception of fetal activity and uterine contractions to provide a marker against which any changes in fetal heart rate or fetal pattern of activity can be assessed."

The authors would like to thank the reviewer for their helpful comments. The discussion has been amended to include the additional issues that must be addressed by continuous fetal monitoring (specifically normal rhythms of fetal behaviour, sleep patterns, responses to contractions and fetal movement patterns). We have also included a sentence to raise the issue that adding an interactive component for mothers perceive contractions (or other symptoms such as pain) would be an important development as such devices move towards the clinical arena. 

Please note, the following figures and supplementary information have been also been edited: 

• Figure 1: 5 studies were added to the boxes titled ‘additional records identified through other sources’, ‘records after duplicates removed’, ‘records screened’, ‘full-text articles assessed for eligibility’ and ‘studies included in qualitative synthesis’. 

• Figure 3: success rates for two studies were added: Verdurmen et al. 2018 (42.9%) and Noben et al. 2019 (73.3%).

• S1 File: page numbers for relevant sections were altered where necessary

---

## [Decision Letter · Decision Letter 1]

14 Oct 2020

PONE-D-20-20354R1

A systematic scoping review to identify the design and assess the performance of devices for antenatal continuous fetal monitoring

PLOS ONE

Dear Dr. Tamber,

Thank you for submitting your manuscript to PLOS ONE. After careful consideration, we feel that it has merit but does not fully meet PLOS ONE’s publication criteria as it currently stands. Therefore, we invite you to submit a revised version of the manuscript that addresses the minor points raised during the review process by one of the reviewers.

We look forward to receiving your revised manuscript.

Kind regards,

Martin G Frasch, MD, PhD

Academic Editor

PLOS ONE

Reviewers' comments:

Reviewer's Responses to Questions

**Comments to the Author**

1. If the authors have adequately addressed your comments raised in a previous round of review and you feel that this manuscript is now acceptable for publication, you may indicate that here to bypass the “Comments to the Author” section, enter your conflict of interest statement in the “Confidential to Editor” section, and submit your "Accept" recommendation.

Reviewer #1: (No Response)

Reviewer #2: All comments have been addressed

Reviewer #3: All comments have been addressed

2. Is the manuscript technically sound, and do the data support the conclusions?

Reviewer #1: Yes

Reviewer #2: Yes

Reviewer #3: Yes

3. Has the statistical analysis been performed appropriately and rigorously? 

Reviewer #1: Yes

Reviewer #2: Yes

Reviewer #3: Yes

4. Have the authors made all data underlying the findings in their manuscript fully available?

Reviewer #1: Yes

Reviewer #2: Yes

Reviewer #3: Yes

5. Is the manuscript presented in an intelligible fashion and written in standard English?

Reviewer #1: Yes

Reviewer #2: Yes

Reviewer #3: Yes

6. Review Comments to the Author

Reviewer #1: The authors have processed comments from the reviewers and in my opinion the paper has been improved quite a lot. I only have a few small comments left:

In Figure 2, the relative weights in the meta-analysis are given. Do these weights consider the fact that different studies might have shared the same data. Specifically, the studies by Graatsma might have been based on partly the same data. If this was the case, and I can understand if the authors don’t know this because I am not sure whether Graatsma et al. commented on this in their second paper, should the meta-analysis be corrected for repeated data? If appropriate, the authors could discuss this as a (minor) limitation.

Regarding the last paragraph of the conclusion, I agree with the unclear impact of certain maternal and fetal factors and the high variability between reported performances. But with some studies having a very small number of participants, I think the conclusion would benefit from the authors spending a few words on the performances reported in the large studies. If these would show smaller variability, the variability in the entire review might be (partly and perhaps unfairly) ascribed to the small studies. I know from experience that performance of fECG recordings improves with more recordings being done; clinical users seem to experience a learning curve in the preparation of the recording/placement of the electrodes. So one could expect that in small studies this learning curve plays a larger role than in large studies.

I also agree with the authors that based on their studied literature, CFM cannot yet be recommended. However, could the authors also share their opinion on current alternatives? If there are no alternative methods for CFM, would it be better to not monitor at all or e.g. combine CFM with intermittent traditional CTG?

A few comments on the text:

Line 99: two points (.)

Line 214: five different instead of fivedifferent

Line 278: remove “of”

Line 413: section should start on new line

Line 512: authors use the word successfully twice. I think the second usage is not needed

Line 529/530: impact of respiratory artefacts instead of impact respiratory artefacts

Line 581: appear instead of appears?

Reviewer #2: All issues have been addressed. There are no further issues open. The manuscript is acceptable for publication.

Reviewer #3: See previous comment on this article. I believe the authors have responded appropriately to my comments. No additional comments.

7. PLOS authors have the option to publish the peer review history of their article (what does this mean?). If published, this will include your full peer review and any attached files.

Reviewer #1: No

Reviewer #2: No

Reviewer #3: No

---

## [Author Response · Author response to Decision Letter 1]

8 Nov 2020

The authors would like to thank the editor and reviewers for their comments on our manuscript, particularly during a time when clinicians and academics have been very busy adapting to the challenges posed by COVID-19. We have addressed each of the comments in detail below and amended our manuscript accordingly. The changes and responses are shown in blue font. 

"Reviewer #1: 

The authors have processed comments from the reviewers and in my opinion the paper has been improved quite a lot. I only have a few small comments left:

In Figure 2, the relative weights in the meta-analysis are given. Do these weights consider the fact that different studies might have shared the same data. Specifically, the studies by Graatsma might have been based on partly the same data. If this was the case, and I can understand if the authors don’t know this because I am not sure whether Graatsma et al. commented on this in their second paper, should the meta-analysis be corrected for repeated data? If appropriate, the authors could discuss this as a (minor) limitation."

The authors would like to thank the reviewer for their helpful comments. The relative weights in the meta-analysis do not consider the fact that some studies may share data; this was because the authors of the relevant papers did not explicitly state whether the results/data had been previously published in other papers. Therefore, as per your suggestion, we have discussed this as an additional limitation of our analysis.

"Regarding the last paragraph of the conclusion, I agree with the unclear impact of certain maternal and fetal factors and the high variability between reported performances. But with some studies having a very small number of participants, I think the conclusion would benefit from the authors spending a few words on the performances reported in the large studies. If these would show smaller variability, the variability in the entire review might be (partly and perhaps unfairly) ascribed to the small studies. I know from experience that performance of fECG recordings improves with more recordings being done; clinical users seem to experience a learning curve in the preparation of the recording/placement of the electrodes. So one could expect that in small studies this learning curve plays a larger role than in large studies."

The authors agree that the apparent performance of the devices could be affected by a learning curve in their use. Therefore, one would expect the variability to be lesser in the larger studies (the sample size for studies is shown in table 1. Only the MONICA AN24 and FMAM devices have sufficient number of studies to determine whether there is a relationship between sample size and reported signal quality. We performed a regression analysis on the Monica AN24 studies which showed no significant relationship between study size and estimated SQ (p=0.53, r2=0.06). We have included this information in the results and discussion sections of the revised manuscript. Unfortunately we could not perform this analysis on the FMAM devices as the device performance was not reported in a similar format. 

"I also agree with the authors that based on their studied literature, CFM cannot yet be recommended. However, could the authors also share their opinion on current alternatives? If there are no alternative methods for CFM, would it be better to not monitor at all or e.g. combine CFM with intermittent traditional CTG?"

The use of intermittent monitoring throughout high-risk pregnancies does undoubtedly provide reassurance to pregnant women, and helps relieve anxiety even if this is only in the short-term; therefore a form of monitoring should be used if there are concerns for fetal wellbeing in pregnancy. Whilst clinical studies are still undergoing to assess the reliability of CFM it is also important that intermittent monitoring continues to take place so we do not deviate from current ‘gold-standard’ forms of practice. The anticipated aim of CFM is to provide a method of long-term monitoring of high-risk fetuses, and this will there alleviate the need for intermittent traditional CTG. We have clarified this point in the discussion.

"A few comments on the text:

• Line 99: two points (.)

• Line 214: five different instead of fivedifferent

• Line 278: remove “of”

• Line 413: section should start on new line

• Line 512: authors use the word successfully twice. I think the second usage is not needed

• Line 529/530: impact of respiratory artefacts instead of impact respiratory artefacts

• Line 581: appear instead of appears?"

We have amended these typographical errors in the re-submitted manuscript.

"Reviewer #2: 

All issues have been addressed. There are no further issues open. The manuscript is acceptable for publication.

Reviewer #3: 

See previous comment on this article. I believe the authors have responded appropriately to my comments. No additional comments."

The authors would like to thank the reviewers for re-reading the submission.

---

## [Decision Letter · Decision Letter 2]

13 Nov 2020

A systematic scoping review to identify the design and assess the performance of devices for antenatal continuous fetal monitoring

PONE-D-20-20354R2

Dear Dr. Tamber,

We’re pleased to inform you that your manuscript has been judged scientifically suitable for publication and will be formally accepted for publication once it meets all outstanding technical requirements.

Kind regards,

Martin G Frasch, MD, PhD

Academic Editor

PLOS ONE

Additional Editor Comments (optional):

Reviewers' comments:

Reviewer's Responses to Questions

**Comments to the Author**

1. If the authors have adequately addressed your comments raised in a previous round of review and you feel that this manuscript is now acceptable for publication, you may indicate that here to bypass the “Comments to the Author” section, enter your conflict of interest statement in the “Confidential to Editor” section, and submit your "Accept" recommendation.

Reviewer #1: All comments have been addressed

2. Is the manuscript technically sound, and do the data support the conclusions?

Reviewer #1: Yes

3. Has the statistical analysis been performed appropriately and rigorously? 

Reviewer #1: Yes

4. Have the authors made all data underlying the findings in their manuscript fully available?

Reviewer #1: Yes

5. Is the manuscript presented in an intelligible fashion and written in standard English?

Reviewer #1: Yes

6. Review Comments to the Author

Reviewer #1: (No Response)

7. PLOS authors have the option to publish the peer review history of their article (what does this mean?). If published, this will include your full peer review and any attached files.

Reviewer #1: No

---

## [Editor Report · Acceptance letter]

19 Nov 2020

PONE-D-20-20354R2 

A systematic scoping review to identify the design and assess the performance of devices for antenatal continuous fetal monitoring 

Dear Dr. Tamber:

I'm pleased to inform you that your manuscript has been deemed suitable for publication in PLOS ONE. Congratulations! Your manuscript is now with our production department. 

Kind regards, 

on behalf of

Dr. Martin G Frasch 

Academic Editor

PLOS ONE